# UNCERTAINTY-AWARE PPG-2-ECG FOR ENHANCED CARDIOVASCULAR DIAGNOSIS USING DIFFUSION MODELS

## ABSTRACT

Analyzing the cardiovascular system condition via Electrocardiography (ECG) is a common and highly effective approach, and it has been practiced and perfected over many decades. ECG sensing is non-invasive and relatively easy to acquire, and yet it is still cumbersome for holter monitoring tests that may span over hours and even days. A possible alternative in this context is Photoplethysmography (PPG): An optically-based signal that measures blood volume fluctuations, as typically sensed by conventional "wearable devices". While PPG presents clear advantages in acquisition, convenience, and cost-effectiveness, ECG provides more comprehensive information, allowing for a more precise detection of heart conditions. This implies that a conversion from PPG to ECG, as recently discussed in the literature, inherently involves an unavoidable level of uncertainty. In this paper we introduce a novel methodology for addressing the PPG-2-ECG conversion, and offer an enhanced classification of cardiovascular conditions using the given PPG, all while taking into account the uncertainties arising from the conversion process. We provide a mathematical justification for our proposed computational approach, and present empirical studies demonstrating its superior performance compared to state-of-the-art baseline methods.

## 1 INTRODUCTION

Cardiovascular diseases are a significant public health problem, affecting millions of people worldwide and remaining a leading cause of mortality (Tsao et al., 2022). The analysis of cardiovascular conditions using Electrocardiography (ECG) signals has emerged as one of the most widely utilized diagnostic tool (Kligfield et al., 2007), and its practice and efficiency have been continuously refined over many decades.

While ECG sensing is non-invasive and relatively easy to acquire, it is time-consuming and requires the expertise of trained professionals with specialized skills in order to ensure accurate diagnosis. Consequently, there has been a surge in the development of numerous contemporary wearable ECG systems in recent decades for digital diagnosis. However, the materials used to deliver a high-quality signal via electrodes can often lead to skin irritation and discomfort during extended usage, thereby limiting the long-term viability of these devices (Zhu et al., 2019).

A possible alternative to ECG sensing and diagnosis is the use of Photoplethysmography (PPG), which is an optically-based non-invasive signal associated with rhythmic changes in blood volume within tissues (Reisner et al., 2008). Unlike ECG, PPG signals are easier to obtain, convenient, and cost-effective, as they are widely available in clinics and hospitals and can be sensed through finger/toe clips. Moreover, the popularity of PPG is increasing with the emergence of wearable devices like smart watches, enabling continuous long-term monitoring without causing skin irritations.

Naturally, the PPG and ECG signals are inter-related, as the timing, amplitude, and shape characteristics of the PPG waveform contain information about the interaction between the heart and the blood vessels. These features of the PPG have been leveraged to measure heart rate, heart rate variability, respiration rate (Karlen et al., 2013), blood oxygen saturation (Aoyagi & Miyasaka, 2002), blood pressure (Payne et al., 2006), and to assess vascular function (Marston, 2002; Allen & Murray, 1993). Despite these capabilities, using PPG-based monitoring during daily activities and light

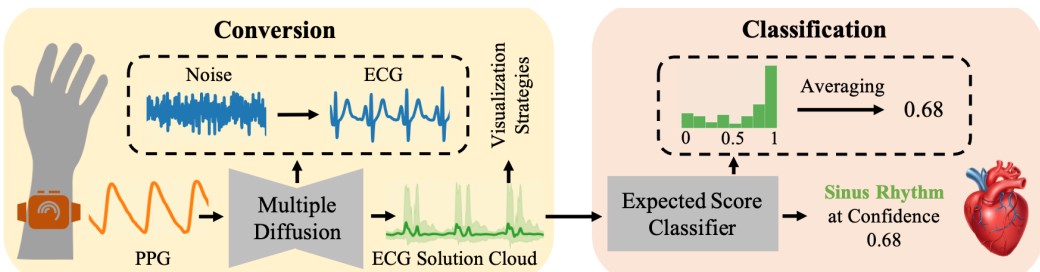

Figure 1: Illustration of our proposed UA-P2E for PPG-2-ECG conversion-classification framework.

physical exercises poses unavoidable uncertainty challenges due to signal inaccuracies and inherent noise, stemming from their indirect nature, compared to ECG signals sourced directly from the heart.

As the use of wearable devices capturing PPG signals is becoming widespread, there is an emerging interest in accurately converting PPG sources into ECG signals (Chiu et al., 2020; Li et al., 2020; Zhu et al., 2021; Sarkar & Etemad, 2021; Vo et al., 2021; Tang et al., 2022; Ho et al., 2022; Tian et al., 2022; Vo et al., 2023; Lan, 2023; Abdelgaber et al., 2023; Hinatsu et al., 2023; Shome et al., 2023). However, such a mapping can be regarded as an ill-posed inverse problem. This implies that multiple ECG solutions may correspond to the same input PPG signal. Nevertheless, previous work in this domain has tended to focus on providing a single ECG solution to the conversion task, ignoring the inherent uncertainty in the conversion and thus failing to capture the complete information contained in the PPG signal. Such studies suggest optimizing neural models either via regularized regression with an $L_1/L_2$ loss (Li et al., 2020; Chiu et al., 2020; Zhu et al., 2021; Tang et al., 2022; Ho et al., 2022; Tian et al., 2022; Lan, 2023; Abdelgaber et al., 2023; Hinatsu et al., 2023) or relying on a generative approach that produces a single solution that approximates a sample from the posterior distribution of ECG given the PPG signal (Sarkar & Etemad, 2021; Vo et al., 2021; 2023; Shome et al., 2023). Here, we argue that both these strategies are lacking, and the conversion in question calls for a different approach.

In this paper, we introduce *Uncertainty-Aware PPG-2-ECG* (UA-P2E), a novel PPG-2-ECG conversion methodology, aimed at enhancing the classification of cardiovascular conditions using ECG signals derived from PPG data. Unlike prior work, our approach aims to address both the uncertainty arising from the conversion task, regarding the spread and variability of the possible ECG solutions given the PPG (Angelopoulos et al., 2022; Belhasin et al., 2023), and also the uncertainty in the classification task, enabling the selection of PPG samples with low classification risk and high confidence (Geifman & El-Yaniv, 2017).

To achieve the above, we leverage a conditional diffusion-based methodology (Sohl-Dickstein et al., 2015; Ho et al., 2020; Song et al., 2020). These generative models have recently emerged as the leading synthesis approach in various tasks (Dhariwal & Nichol, 2021; Yang et al., 2023), and have been shown to enable high-fidelity posterior modeling for inverse problems (Yang et al., 2023; Kawar et al., 2021; 2022; Song et al., 2022; Chung et al., 2022). We utilize this capability to produce a multitude of posterior sampled ECG results that correspond to the input PPG, and provide classification decisions based on this cluster of outcomes. Additionally, we introduce several methods for displaying the resulting ECG candidates for better human interpretability. Figure 1 provides an illustration of our approach to the conversion-classification process.

Our computational methodology is grounded on a mathematical proof showcasing the optimality of the expected classification score derived from our proposed approach for classifying cardiovascular conditions from source PPG signals. This optimality relies on the assumption that we have access to a perfect posterior sampler and ECG classification models. This stands in contrast to the single-solution approach used by prior work, as described above, which necessarily performs worse.

Our work presents a thorough empirical study demonstrating superior performance in classification and uncertainty estimation compared to baseline methods. Our main results are demonstrated on the "Computing in Cardiology" (CinC) dataset (Reyna et al., 2021), which provides the most challenging classification data in cardiology.

In summary, our contributions are the following: (1) We introduce a state-of-the-art diffusion-based methodology for PPG-2-ECG conversion-classification that accounts for the uncertainty in the conversion task, thereby enhancing performance; (2) We provide a mathematical proof that justifies our proposed computational process for the classification task; (3) We offer effective methods for displaying the ECG solutions obtained, enhancing interpretability while taking into account the uncertainty perspective; and (4) We present an empirical study that supports our mathematical proof and demonstrates the superiority of our methodology over baseline strategies.

## 2 RELATED WORK

Naturally, our work is not the first one to consider the conversion of PPG signals to ECG ones. Below we outline the main trends in this line of work, and add a discussion on uncertainty quantification, as we aim to practice in this paper.

**Regression:** Earlier approaches to the PPG-2-ECG conversion problem primarily focused on approximating either the Minimum Mean Absolute Error (MMAE) estimator using $L_1$ loss (Li et al., 2020; Chiu et al., 2020; Ho et al., 2022; Lan, 2023) or the Minimum Mean Squared Error (MMSE) estimator with $L_2$ loss (Zhu et al., 2021; Tang et al., 2022; Tian et al., 2022; Abdelgaber et al., 2023; Hinatsu et al., 2023). These works commonly employed regularization techniques to enhance perceptual outcomes. However, the work reported in Blau & Michaeli (2018) which discovered the distortion-perception tradeoff has shown that when optimizing for distortion (of any kind), perceptual quality is necessarily compromised, thus pushing the solutions in the above methods to signals that are likely to drift out of the valid ECG manifold, thus weakening their classification performance.

**Generation:** The first attempt at PPG-2-ECG conversion using generative models was introduced by the authors of CardioGAN (Sarkar & Etemad, 2021). They employed an adversarial training scheme with an encoder-decoder architecture. Similarly, P2E-WGAN (Vo et al., 2021) utilized conditional Wasserstein GANs for this task. Vo et al. (2023) introduced a variational inference methodology to approximate the posterior distribution with a Gaussian prior assumption. The authors of RDDM (Shome et al., 2023) have first adapted a diffusion-based model that accelerates a DDPM (Ho et al., 2020) diffusion process for this conversion task. However, despite the success of the generative approach, all the above works have chosen to use a single (possibly random) ECG solution from the posterior distribution, thus missing the complete picture of the possibilities of the different ECGs given the PPG signal.

**Uncertainty Quantification:** In the framework of the conversion-classification explored in this paper, uncertainty quantification serves a dual purpose. Firstly, it seeks to characterize the range, spread, and variability of potential solutions for a given input PPG signal (Angelopoulos et al., 2022; Belhasin et al., 2023). Secondly, it also aims at assessing the confidence in the correctness of classification predictions, e.g. via a selective classification (Geifman & El-Yaniv, 2017) scheme, which we adopt in this paper. Building upon principles from these two threads of work, we characterize the diverse solutions that can potentially describe the input, leading to an improved methodology within the conversion-classification framework.

## 3 PROBLEM FORMULATION AND GOALS

Let $X \in \mathcal{X} \subseteq \mathbb{R}^d$ be a random vector representing an ECG signal that follows the prior distribution $\pi(X)$. We assume access only to its observation $Y = h(X) \in \mathcal{Y} \subseteq \mathbb{R}^d$, where $h$ is an unknown, possibly stochastic, and non-invertible function. The observation $Y$ represents a PPG signal that follows the distribution $\pi(Y|X)$. In this paper, our first and primary objective is to provide a comprehensive and interpretable estimation of the ECG signal(s) that corresponds to the given $Y$, focusing on the approximation of the posterior distribution $\pi(X|Y)$. Prior work in this field (see Section 2) adopted a *single-solution* approach for the conversion task, focusing on estimating a single reconstructed ECG solution from $\pi(X|Y)$, typically achieved through regression or generation models.

Here, we propose a generalization of this conversion approach using a conditional stochastic generative model, denoted as $g : \mathcal{Y} \times \mathcal{Z} \to \mathcal{X}$ s.t. $g(Y, Z) = \hat{X} \sim \hat{\pi}(X|Y)$, where $Z \in \mathcal{Z}$ denotes

a stochastic component that typically follows a normal distribution $\pi(Z) := \mathcal{N}(0, \mathbf{I})$, and $\hat{X}$ represents a potential ECG solution derived as a sample from the approximate posterior distribution $\hat{\pi}(X|Y)$. In this context, a single ECG solution may be derived by fixing $Z = z$ in $g(Y, Z)$, or by marginalizing over the random seed, i.e., $\mathbb{E}_{z \sim \pi(Z)}[g(Y, Z)]$. In contrast, allowing $Z$ to be random may provide a more comprehensive set of solutions to the inverse problem, exposing the inherent uncertainty within this conversion task. Our prime goal in this work is therefore to train the stochastic conversion model $g$ using a training data consisting of many (ECG, PPG) signal pairs, denoted by $\{(X_i, Y_i)\}_{i=1}^n$.

A second objective in this work is classification, enabling diagnosis of various cardiovascular conditions via the incoming PPG signals. Consider a binary classification problem where the input is an observation $Y \in \mathcal{Y}$ with a corresponding binary label $C \in \{0, 1\}$ that follows a conditional distribution $\pi(C|Y)$. In this classification setting, we aim to model a classifier $f_\mathcal{Y} : \mathcal{Y} \to \mathbb{R}^+$ for approximating the posterior distribution, denoted as $\hat{\pi}(C|Y)$.

Addressing this classification task, one might be tempted to adopt a straightforward and naive strategy, of training directly on data that consists of PPG signals and their associated labels. Nevertheless, appropriate cardiovascular labels linked with PPG signals are relatively rare and hard to acquire. An alternative is to use labels that were produced for ECG signals, $X$, and tie these to their matched PPG, $Y$. However, as the mapping between PPG and ECG is not injective, this implies that other ECG signals (possibly with a different label) could have led to the same PPG. As a consequence, these labels are necessarily noisy, impairing the learning task.

A presumably better alternative is to operate in the reconstructed ECG domain, hoping for improved performance. Indeed, the classification proposed in prior work operates on the single estimated ECG $\hat{X}$, approximating this way the posterior $\pi(C|\hat{X})$. This is achieved using a classifier $f_\mathcal{X} : \mathcal{X} \to \mathbb{R}^+$, trained over data of the form $\{(X_i, C_i)\}_{i=1}^n$, where $X_i$ represents a grounded ECG signal and $C_i$ denotes its associating label. As we rely now on labels that do match their corresponding signals, this may seem like a better approach. However, the same flaw as above applies here – the single-produced ECG, $\hat{X}$, captures a partial truth about the given PPG, thus weakening the classification. Here is a formal definition of this classification strategy, as practiced in prior work:

**Definition 3.1** (Single Score Classifier). *We define a stochastic classifier $f_\mathcal{Y}$ as a Single Score Classifier (SSC) if it is given by*

$$f_\mathcal{Y}(Y) = f_\mathcal{X}(\mathbb{E}_{Z' \sim \pi(Z')}[g(Y, Z')]).$$

When choosing $\pi(Z') = \pi(Z)$, the term $\mathbb{E}_{Z' \sim \pi(Z')}[g(Y, Z')]$ becomes the MMSE estimate of the ECG, and when $\pi(Z') = \delta(Z - W)$ for $W \sim \pi(Z)$, we get a single sample from the approximate posterior distribution, $\hat{\pi}(X|Y)$.

Following the data processing inequality (Beaudry & Renner, 2011), any transformation $\hat{X} = \mathbb{E}_{Z' \sim \pi(Z')}[g(Y, Z')]$ satisfies the inequality

$$I(Y; C) \geq I(\hat{X}; C), \tag{1}$$

where $I(U; V)$ denotes the mutual information between the random variables $U$ and $V$. The above inequality implies that the classification information contained within a single reconstructed ECG signal $\hat{X}$, as utilized by prior work, is always weakly lower than the classification information contained within the original PPG signal $Y$. This suggests that despite the transformation from PPG to ECG, there may be some loss or reduction in the discriminative power for classification tasks due to information processing constraints.

To address this challenge, we adopt a *multi-solution* formulation approach that takes into account all possible ECG solutions that can be derived from the PPG signal $Y$. Below is the formal definition of our classification strategy.

**Definition 3.2** (Expected Score Classifier). *We define a classifier $f_\mathcal{Y}$ as an Expected Score Classifier (ESC) if it is given by*

$$f_\mathcal{Y}(Y) = \mathbb{E}_{Z \sim \pi(Z)}\left[f_\mathcal{X}(g(Y, Z))\right].$$

Intuitively, this strategy averages the classification scores of posterior samples that emerge from the posterior distribution $\pi(X|Y)$, rather than averaging the samples themselves. As the following Theorem claims, this classification approach is optimal.

**Theorem 3.1** (Optimality of the Expected Score Classifier). *Consider the Markovian dependency chain* $Y \rightarrow X \rightarrow C$, *implying that* $\pi(C|X,Y) = \pi(C|X)$, *and assume the following:*

1. *The conversion model $g$ is a posterior sampler, i.e., $g(Y,Z) = \hat{X} \sim \pi(X|Y)$; and*

2. *The classification model $f_{\mathcal{X}}$ is optimal by satisfying $f_{\mathcal{X}}(X) = \pi(C = 1|X)$,*

*Then the obtained classification is optimal, satisfying $f_{\mathcal{Y}}(Y) = \pi(C = 1|Y)$.*

Supported by Equation (1), Theorem 3.1 implies that our approach utilizes the full extent of the information within the observation $Y$, and is therefore optimal. In particular, this demonstrates that our approach is superior to the single-solution classification strategy (Definition 3.1), which has been adopted by prior work. See Appendix A for the proof of the above claim, along with an extension showing the ESC optimality in practical settings.

# 4 UA-P2E: UNCERTAINTY-AWARE PPG-2-ECG

We now turn to introduce *Uncertainty-Aware PPG-2-ECG* (UA-P2E), our methodology for PPG-2-ECG conversion, aiming to provide a comprehensive understanding of the various ECG solutions derived from a given PPG signal. UA-P2E may assist medical professionals in analyzing diverse ECG signal variations resulting from the conversion process. Additionally, our method can be leveraged to enhance cardiovascular diagnostic accuracy and confidence assessments, facilitating more informed clinical decisions.

While our proposed approach is applicable using any conditional stochastic generative estimator, denoted as $g : \mathcal{Y} \times \mathcal{Z} \rightarrow \mathcal{X}$, we focus in this work on conditional diffusion-based models (Sohl-Dickstein et al., 2015; Ho et al., 2020; Song et al., 2020), which have emerged as the leading synthesis approach in various domains (Dhariwal & Nichol, 2021; Yang et al., 2023). Our diffusion process is trained and evaluated on raw signal data of fixed-length $Y, X \in \mathbb{R}^d$ PPG and ECG signal pairs, sampled at 125Hz, thus covering $d/125$ seconds.[1] The central part of the diffusion process involves a conditional denoiser, denoted as $\epsilon(X_t, Y, t)$. This denoiser is trained to predict the noise in $X_t$, which is a combination of $X_0$ (representing a clean instance) and white additive Gaussian noise at a noise level denoted by $t$. In this conditional process, the denoiser leverages the PPG signal $Y$ and the noise level $t$ as supplementary information. Training $\epsilon$ is based on pairs of $(X, Y)$ signals - a clean ECG and its corresponding PPG. After training, we apply multiple diffusion operations on a given PPG signal $Y \in \mathcal{Y}$ to draw potential ECG solutions $\{\hat{X}_i\}_{i=1}^K$, where each $\hat{X}_i \sim \hat{\pi}(X|Y)$ is a sample from the approximated posterior distribution of the ECG given the PPG signal.

In Section 4.1 we describe our enhanced classification procedure that relies on the above posterior sampler, denoted by $g$. Additionally, in Section 4.2 we discuss how to improve the quantification of uncertainty in classification, aiming to enhance confidence in predictions. Lastly, in Section 4.3, we offer methods to visualize the cloud of ECG solutions $\{\hat{X}_i\}_{i=1}^K$. This visualization aims at enhancing interpretability for medical professionals, allowing for a better understanding of the various ECG signals that emerge from a given PPG signal.

## 4.1 UA-P2E: ENHANCED CLASSIFICATION

The methodology for UA-P2E's enhanced classification consists of three phases. The first involves training the probabilistic classification model, denoted as $f_{\mathcal{X}} : \mathcal{X} \rightarrow \mathbb{R}^+$, using ECG signals and their associated labels. $f_{\mathcal{X}}$ can be optimized through the commonly used Cross-Entropy loss function, producing an output that has a probabilistic interpretation. In the second phase, each potential posterior sample $\hat{X}_i$ from the set $\{\hat{X}_i\}_{i=1}^K$ (derived by applying $g$ to a given $Y$) is evaluated by this classifier, extracting an output score $f_{\mathcal{X}}(\hat{X}_i)$, being an estimated probability of belonging to the positive class ($C = 1$). In the final phase, the average classification score, $\frac{1}{K}\sum_{i=1}^K f_{\mathcal{X}}(\hat{X}_i)$, is computed, and compared to a decision threshold $t \in \mathbb{R}^+$. The complete workflow of our proposed method to enhance classification is detailed in Algorithm 1.

---

[1]in our experiments $d = 1024$ and thus the signals cover $\approx 8$ seconds.

---

**Algorithm 1** UA-P2E for Classification Framework

---

**Require:** Observation $Y \in \mathcal{Y}$. Pretrained conditional stochastic generative model $g : \mathcal{Y} \times \mathcal{Z} \to \mathcal{X}$. Pretrained classification model $f_{\mathcal{X}} : \mathcal{X} \to \mathbb{R}^+$. Number of samples $K \in \mathbb{N}$. Decision threshold $t \in \mathbb{R}^+$.
**Ensure:** Classification decision $\hat{C} \in \{0, 1\}$, and its positive score $f_{\mathcal{Y}}(Y) \in \mathbb{R}^+$.
1: **for** $i = 1$ to $K$ **do**                                                    ▷ Apply the conversion process
2:    Draw $Z_i \sim \mathcal{N}(0, 1)$ and compute $\hat{X}_i \leftarrow g(Y, Z_i)$.
3: **end for**
4: $f_{\mathcal{Y}}(Y) \leftarrow \frac{1}{K} \sum_{i=1}^{K} f_{\mathcal{X}}(\hat{X}_i)$                    ▷ Approximate the ESC (Definition 3.2)
5: **if** $f_{\mathcal{Y}}(Y) > t$ **then** $\hat{C} \leftarrow 1$ **else** $\hat{C} \leftarrow 0$.

---

Algorithm 1 seeks to approximate the ESC (Definition 3.2), which has been proven optimal in Theorem 3.1.

## 4.2 UA-P2E: Uncertainty Quantification

The proposed PPG-2-ECG conversion has two layers of uncertainty awareness, as described herein. The first takes into account the inherent uncertainty that emerges from the ill-posed conversion itself, quantifying the spread and variability of the possible ECG candidates that derive from the PPG inputs. In Appendix C, we present empirical evidence illustrating the uncertainty from this perspective. The second uncertainty layer leverages the above property for providing classification predictions with a better confidence in their correctness. This serves a selective classification (Geifman & El-Yaniv, 2017) scheme, in which a carefully chosen subset of the test PPG data is selected by their calculated confidence, exhibiting improved accuracy.

Selective classification (Geifman & El-Yaniv, 2017) enables handling cases in which classifying every PPG signal within a test set is challenging due to uncertainty or data corruption. This technique allows for abstaining from making predictions for signals with low confidence, thus reducing classification errors on the remaining data. A confidence function, defined as $\kappa : \mathcal{Y} \to \mathbb{R}^+$, serves as the engine for this task by predicting the correctness of predictions. One common choice of this confidence function $\kappa$ is to use the maximal softmax score of a classifier. In a similar spirit, in this work we set $\kappa(Y)$ to take the value $\kappa(Y) = \max\{1 - f_{\mathcal{Y}}(Y), f_{\mathcal{Y}}(Y)\}$, where $f_{\mathcal{Y}}(Y)$ approximates the ESC (Definition 3.2), as summarized in Algorithm 1.

In addition to the above, we may consider applying a calibration scheme on a heldout set, denoted as $\{(Y_i, C_i)\}_{i=1}^{m}$, to determine a threshold $\lambda \in \mathbb{R}^+$ for $\kappa(Y)$ that defines the reliable test PPG signals whose classification error is statistically guaranteed to be lower than a user-defined risk level $\alpha \in (0, 1) \ll 1$. Such a guarantee has the following form:

$$\mathbb{P} \left( \frac{\mathbb{E}_{\pi(Y,C)} \left[ \mathbf{1}\{\hat{C} \neq C\} \mathbf{1}\{\kappa(Y) > \lambda\} \right]}{\mathbb{E}_{\pi(Y,C)} \left[ \mathbf{1}\{\kappa(Y) > \lambda\} \right]} < \alpha \right) > 1 - \delta, \tag{2}$$

where $\delta \in (0, 1) \ll 1$ is a user-defined calibration error. Intuitively, by setting $\alpha = \delta = 0.1$, the above implies that the classification error among selected PPG signals, whose confidence scores are greater than $\lambda$, will be lower than $\alpha \ (= 10\%)$ with probability of $1 - \delta \ (= 90\%)$. In Appendix B, we describe how to calibrate $\lambda$ to provide the guarantee stated in Equation (2).

## 4.3 Visualizing UA-P2E's ECG Output(s)

A primary objective of UA-P2E is the PPG-2-ECG conversion, and a natural question to pose is what ECG signal to present to medical professionals after such a conversion. Due to the stochastic nature of our estimation approach, this question becomes even more intricate, as we have multitude of candidate ECG signals, all valid from a probabilistic point of view. We emphasize, however, that there is no "correct" answer to the question of the ECG result to present, and thus the options discussed hereafter can be considered as possibilities to be chosen by the human observer.

Note that although a single ECG solution represents only a partial truth that refers to the given PPG signal (see Section 3), here we aim to present such a single ECG that encapsulates the main information conveyed by the given PPG signal. In terms of interpretability, we assume that professionals

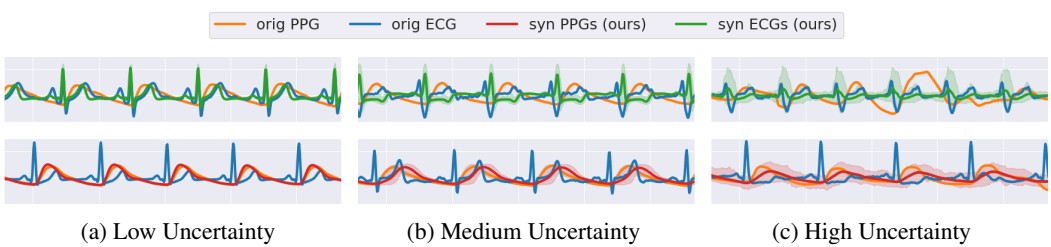

|  (a) Low Uncertainty  |  (b) Medium Uncertainty  |  (c) High Uncertainty  |

Figure 2: **MIMIC-III Results**: Diffusion-based multiple solutions in the temporal domain. Each temporal point displays an interval containing 95% of our ECG/PPG synthetic values. The top row depicts the ECG solution cloud (GREEN) resulting from the PPG-2-ECG conversion, while the bottom row shows the PPG solution cloud (RED) from the reverse conversion of ECG-2-PPG.

might prefer receiving a single solution that closely aligns with the underlying medical information within the PPG signal.

To address this challenge, while acknowledging the inherent uncertainty in converting PPG-2-ECG from a classification perspective, we first apply our diagnostic approach as described in Section 4.1, which involves analyzing multiple ECG solutions. Consequently, we propose presenting a single ECG that corresponds to the diagnostic findings, specifically tied to the histogram of ECG solutions whose classification aligns with the classification obtained using the ESC (Definition 3.2) approximation. Formally, these ECG solutions are derived from $\{f_{\mathcal{X}}(\hat{X}_i)\}_{i=1}^k$, where $\hat{X}_i = g(Y, Z_i) \sim \hat{\pi}(X|Y)$ and $\mathbf{1}\{f_{\mathcal{X}}(\hat{X}_i) > t\} = \mathbf{1}\{f_{\mathcal{Y}}(Y) > t\}$, with $t \in \mathbb{R}^+$ representing the decision threshold. We may offer to use one of the following strategies:

**Most Likely Score ECG:** Present an ECG sample $\hat{X}^*$ that matches the most likely score. i.e.,

$$\hat{X}^* \in \arg \max_{b \in \{I_1, I_2, \ldots, I_B\}} \text{KDE}_{\{f_{\mathcal{X}}(\hat{X}_i)\}_{i=1}^k}(b) \, , \tag{3}$$

where $b \in \{I_1, I_2, \ldots, I_B\}$ denotes the bin intervals of the KDE function of $\{f_{\mathcal{X}}(\hat{X}_i)\}_{i=1}^k$.

**Expected Score ECG:** Present an ECG sample $\hat{X}^*$ that best matches the average score, i.e.,

$$\hat{X}^* = \arg \min_{\hat{X} \in \{\hat{X}_i\}_{i=1}^k} \left| f_{\mathcal{X}}(\hat{X}) - \frac{1}{k} \sum_{i=1}^k f_{\mathcal{X}}(\hat{X}_i) \right| \, . \tag{4}$$

**Min/Max Score ECG:** Present an ECG sample $\hat{X}^*$ that best represents the classification, i.e.,

$$\hat{X}^* = \arg \max_{\hat{X} \in \{\hat{X}_i\}_{i=1}^k} f_{\mathcal{X}}(\hat{X}) \ \textbf{if} \ f_{\mathcal{Y}}(Y) = 1 \ \textbf{else} \ \arg \min_{\hat{X} \in \{\hat{X}_i\}_{i=1}^k} f_{\mathcal{X}}(\hat{X}) \, . \tag{5}$$

In Section 5.3, we provide empirical results regarding the quality of the three visualization approaches mentioned above. These results suggest that the most likely score ECG, as defined in Equation (3), provides the highest quality.

## 5 EMPIRICAL STUDY

This section presents a comprehensive empirical study of our proposed method, UA-P2E, for converting PPG signals to ECG ones (lead II), as detailed in Section 4. All experiments described hereafter are conducted using three random seeds, and the results shown are the average of their outcomes. We employ this conversion technique with two prime objectives in mind: (1) Providing an interpretable estimation of the ECG signals, denoted as $\hat{X}$, that correspond to the given PPG $Y$, while targeting the approximation of samples from the posterior distribution $\pi(X|Y)$; and (2) Providing accurate classification and confidence assessment for a series of cardiovascular conditions. In both objectives, the presented results showcase the superiority of the proposed paradigm across all conducted experiments.

|  |  | RMSE ↓ | 1-FD ↓ | 100-FD ↓ |
|---|---|---|---|---|
| Baseline Methods | CardioGAN | 0.54 | 99.0 | - |
|  | RDDM (T=50) | **0.22** | 6.71 | - |
| **Our Method:** | MMSE-Approx | **0.2222 ± 0.0000** | 11.527 ± 0.0179 | - |
| **UA-P2E** | DDIM (T=25) | 0.3031 ± 0.0000 | 2.8671 ± 0.0034 | 2.7873 ± 0.0007 |
|  | DDIM (T=50) | 0.2956 ± 0.0000 | 0.5194 ± 0.0015 | 0.4418 ± 0.0007 |
|  | DDIM (T=100) | 0.2939 ± 0.0001 | **0.3198 ± 0.0020** | **0.2379 ± 0.0005** |

Table 1: **MIMIC-III Results**: Quality assessment of our diffusion-based conversion model compared to state-of-the-art baseline methods: CardioGAN (Sarkar & Etemad, 2021) and RDDM (Shome et al., 2023). The results show the mean metric estimates and their standard errors across the three seeds.

Unlike prior work, this paper aims to address the uncertainty in the conversion of PPG to ECG signals, and this is obtained by leveraging a diffusion-based conversion methodology. More specifically, we adopt the DDIM framework (Song et al., 2020), aiming to reduce the number of iterations (in our experiments we applied $T = 100$ iterations) required for the sampling procedure, and thus increasing the efficiency of our conversion algorithm. The diffusion denoising model is trained using the MIMIC-III matched waveform database (Johnson et al., 2016), which is one of the most extensive dataset for paired PPG and ECG signals. For details on supplementary analysis of overfitting during the training process, we refer the reader to Appendix D.

Our strategy relies on the core ability of generative models to obtain an effective sampling from the posterior distribution $\hat{\pi}(X|Y)$, where $Y$ stands for the given PPG and $X$ for its corresponding ECG. Figure 2 (top row) illustrates typical sampling outcomes of our diffusion-based conversion model, showcasing the diverse ECG signals generated from different PPG inputs, each exhibiting varying levels of uncertainty, as expressed by the variability of the samples obtained.

As for the classification objective, we train and evaluate the classification model, denoted as $f_{\mathcal{X}}$, using the 'Computing in Cardiology" (CinC) (Reyna et al., 2021) dataset, containing multi-labeled ECG signals of 11 cardiovascular conditions [2] that are detectable by lead II ECG signals. Interestingly, our experiments rely in part on a reversed diffusion-based conversion model, designed to sample from the likelihood distribution $\hat{\pi}(Y|X)$, i.e. converting ECG (lead II) to PPG. This reversed sampling is utilized in order to complete the CinC (Reyna et al., 2021) dataset used in our tests, enabling an analysis of challenging cardiac classification tasks. Figure 2 (bottom row) presents typical sampling outcomes of the reversed conversion model. For further details on experimental details, including the datasets used for training and evaluating the conversion and classification models, along with details on data pre-processing and training schemes, we refer the reader to Appendix E.

## 5.1 DIFFUSION-BASED CONVERSION: QUALITY ASSESSMENT

We begin by assessing the quality of our diffusion-based PPG-2-ECG conversion model using the MIMIC-III (Johnson et al., 2016) dataset, showcasing its superiority over state-of-the-art baseline methods. Here, we examine the quality of $\{\hat{X}_i\}_{i=1}^K$ with $K = 100$, sampled from $\hat{\pi}(X|Y)$.

Table 1 presents the conversion performance results, evaluating the quality of the obtained ECG signals. we provide a comparison with two state-of-the-art methods, CardioGAN (Sarkar & Etemad, 2021) and RDDM (Shome et al., 2023), using their reported metrics. More specifically, we assess the conversion quality through RMSE calculations between the generated ECG(s) and the ground truth signals. Additionally, we employ the Fréchet Distance (FD) to evaluate the perceptual quality of the produced signals, considering either a single ECG sample from each PPG (1-FD) or 100 samples from each (100-FD). The results in Table 1 demonstrate the superiority of our conversion model. Observe that by averaging the $K = 100$ samples, we obtain an approximation of the MMSE estimator, yielding a comparable performance to the reported result in Shome et al. (2023). When considering the individual ECG solutions for each PPG, the RMSE values increases, as expected (Blau & Michaeli, 2018). In terms of perceptual quality, the single-solution and multi-solution approaches

---

[2]The following cardiovascular conditions are considered: (1) Ventricular premature beats, (2) Atrial flutter, (3) Sinus rhythm, (4) Atrial fibrillation, (5) Supraventricular premature beats, (6) Bradycardia, (7) Pacing rhythm, (8) Sinus tachycardia, (9) 1st degree AV block, (10) Sinus bradycardia, and (11) Sinus arrhythmia.

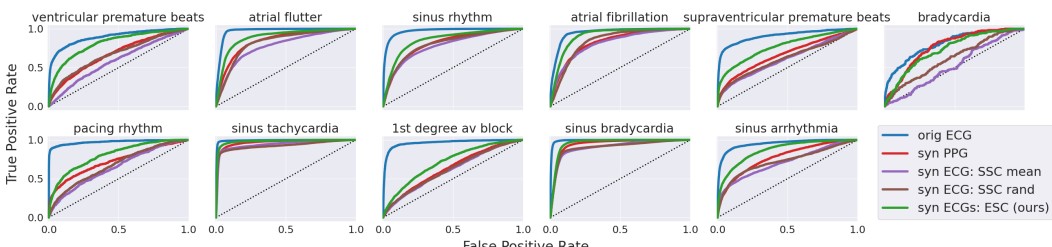

Figure 3: **CinC Results**: ROC curves for the classification performance of various strategies (see Appendix F), showcasing the superiority of our approach. Higher curves indicate better performance.

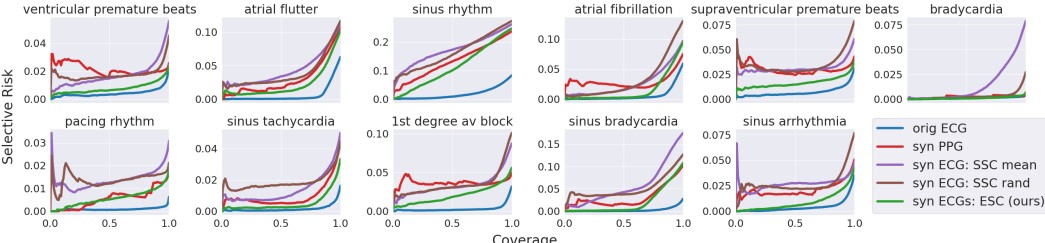

Figure 4: **CinC Results**: Risk-Coverage curves (Geifman & El-Yaniv, 2017) for the uncertainty quantification performance of various strategies, showcasing the superiority of our approach. Lower is better performance.

show pronounced improvement in FD compared to the baselines. Note that computing the FD on multiple solutions helps in improving further the signals' quality.

## 5.2 CLASSIFYING CARDIOVASCULAR CONDITIONS

To demonstrate the benefits of our diffusion-based synthesized ECGs, we present classification results on two datasets, MIMIC-III (Johnson et al., 2016) and CinC (Reyna et al., 2021), analyzing the classification performance across a series of 11 cardiovascular conditions mentioned earlier. For both datasets, we compare our approach to classification strategies utilized by prior work, varied by the choice of $\pi(Z')$ in Definition 3.1, which refers to as the SSC. Specifically, we examine the classification performance of the MMSE estimate and a random single sample from the approximated posterior distribution, $\hat{\pi}(X|Y)$. For more details on the classification strategies considered, refer to Appendix F.

Note that MIMIC-III lacks cardiovascular labels, and thus cannot be utilized as-is for classification. As a remedy, we apply our experiments using the CinC dataset. However, while CinC contains ECG signals (all leads) and their corresponding labels, it does not include PPG signals, and as such it cannot serve our overall PPG classification goals. We resolve this difficulty by generating synthetic PPG signals from the given ECG ones, using a reversed diffusion-based methodology. The idea is to sample valid and random PPG signals that correspond to their ECG (lead II) counterparts, this way augmenting CinC to form a complete suite. In Appendix G we provide empirical evidence for the validity of our reversal conversion model of ECG-2-PPG in comparison with the original PPG signal.

We now turn to describe the main classification results of cardiovascular conditions, referring to the augmented CinC dataset, and using the synthesized PPG signals. Figure 3 presents ROC curves for the classification strategies considered, and Figure 4 completes this description by presenting risk-coverage curves (Geifman & El-Yaniv, 2017), which evaluate the performance of different confidence functions, denoted as $\kappa$, for PPG signals in regard to uncertainty quantification. For supplementary results, refer to Appendix H.

As can be seen from these two figures, our approach (GREEN) demonstrates superior performance over all the alternatives, apart from the direct ECG (BLUE) classification, which serves as an upper-bound for the achievable performance (see Appendix I).

| Visualization Strategy | Signal Domain | | Embedding Domain | |
|---|---|---|---|---|
| | RMSE ↓ | 1-FD ↓ | RMSE ↓ | 1-FD ↓ |
| Min/Max Score ECG (5) | $0.3130 \pm 0.0004$ | $1.3113 \pm 0.0302$ | $0.3079 \pm 0.0037$ | $16.676 \pm 0.7135$ |
| Expected Score ECG (4) | $0.3119 \pm 0.0004$ | $1.2820 \pm 0.0144$ | $0.3091 \pm 0.0044$ | $13.172 \pm 0.2228$ |
| Most Likely Score ECG (3) | $\mathbf{0.3105 \pm 0.0004}$ | $\mathbf{1.2726 \pm 0.0101}$ | $\mathbf{0.3040 \pm 0.0043}$ | $\mathbf{10.491 \pm 0.1879}$ |

Table 2: **CinC Results**: Quality assessment of the three proposed ECG visualization strategies, as described in Section 4.3. The results show the mean metric estimates and their standard errors across the three seeds and demonstrate that the Most Likely Score ECG strategy is the best among the strategies in terms of perceptual quality.

Theorem 3.1 may be interpreted as claiming that our approach is expected to align with that of classifying the original PPG signals directly (ORANGE), under optimal conditions. However, we observe that our approach achieves better performance. We argue that this gap is due to the erroneous labels used in the direct classification: Note that these labels refer to the original ECG signals, and do not necessarily match the corresponding PPG, which may correspond to other valid ECG signals.

Still referring to the results in these two figures, it is evident that prior work which takes advantage of either the classification strategies of the mean estimate (PURPLE) or a random ECG sample (BROWN), exhibits a weaker performance compared to our approach. These findings strengthen our claim that ignoring the uncertainty that emerges from the conversion task might impair performance.

## 5.3 Which ECG Solution to Deliver to Medical Professionals?

In this section, we examine the ECG visualization strategies, as discussed in Section 4.3, over the CinC dataset. By comparing our proposed visualization strategies, we aim to provide empirical insights on how to choose between them. As already mentioned in Section 4.3, there is no "correct" choice for the visualization strategy, and it should be considered by the medical user. Table 2 presents the comparative results of the three proposed strategies: Min/Max Score ECG, Expected Score ECG, and Most Likely Score ECG. This table brings RMSE results for the signals chosen and FD values for their distributions, all compared with their ECG origins. The three strategies are evaluated across two domains, first, in the signal domain, and second, in the embedding domain using our classification model, denoted as $f_{\mathcal{X}}$. As can be seen from the table, while all three considered approaches are good performing in these measures, the "Most Likely Score ECG" seem to be the best, both in RMSE and FD.

## 6 Concluding Remarks

This paper presents Uncertainty-Aware PPG-2-ECG (UA-P2E), a novel approach that addresses the inherent uncertainty in the PPG-2-ECG task. By incorporating the potential spread and variability of derived ECG signals from PPG inputs through sampling from a diffusion-based model, we achieve state-of-the-art performance in the conversion task. In addition, we show that our approach enhances cardiovascular classification, through the approximation of our proposed Expected Score Classifier (ESC), which averages classification scores of the ECG solution cloud for each input PPG signal. This classification strategy is proven to be optimal, and demonstrated to perform better than baseline methods in practice. Finally, we present ECG visualization methods, designed to enhance interpretability for medical professionals while considering the multitude ECG solutions associated with each PPG signal.

Our work presents rich empirical results demonstrating the superior classification performance of our approach compared to direct classification methods operating on given PPG signals. However, these results are partially reliant on synthetic PPG signals, for which we provide empirical validation, though further exploration is required to study additional cardiovascular conditions and the potential risk of hallucinations in the signal domain. Additionally, although the motivation for using PPG-2-ECG is based on the widespread use of wearable devices, no database containing signals from wearable settings was utilized due to the lack of publicly available data; instead, data from hospital facilities were used. Future studies in these areas will require additional data collection.

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

# A  OPTIMALITY OF THE EXPECTED SCORE CLASSIFIER

This section provides theoretical results proving the optimality of the Expected Score Classifier (ESC), defined in Definition 3.2.

## A.1  OPTIMAL SETTINGS

We begin by presenting a proof for Theorem 3.1 in optimal settings:

Consider the Markovian dependency chain $Y \to X \to C$, implying that $\pi(C|X, Y) = \pi(C|X)$, and recalling the assumptions of Theorem 3.1:

1. The conversion model $g$ is a posterior sampler, i.e., $g(Y, Z) = \hat{X} \sim \pi(X|Y)$; and

2. The classification model $f_{\mathcal{X}}$ is optimal by satisfying $f_{\mathcal{X}}(X) = \pi(C = 1|X)$.

By definition, the ESC is written as

$$f_{\mathcal{Y}}(Y) = \mathbb{E}_{Z \sim \pi(Z)}\left[f_{\mathcal{X}}(g(Y, Z))\right] = \int f_{\mathcal{X}}\Big(g(Y, Z = z)\Big)\pi(Z = z)dz.$$

Next, we make a change of variables $\hat{X} = g(Y, Z)$. This gives us

$$f_{\mathcal{Y}}(Y) = \int f_{\mathcal{X}}\Big(g(Y, Z = z)\Big)\pi(Z = z)dz$$

$$= \int f_{\mathcal{X}}(\hat{X} = \hat{x})\pi(\hat{X} = \hat{x}|Y)d\hat{x}$$

$$= \mathbb{E}_{\hat{X} \sim \pi(X|Y)}\left[f_{\mathcal{X}}(\hat{X})\right],$$

where we rely on the assumption that $g$ is a posterior sampler, namely, $g(Y, Z) \sim \pi(X|Y)$. Therefore, for any $y \in \mathcal{Y}$, it follows from the optimality of $f_{\mathcal{X}}$ that

$$f_{\mathcal{Y}}(Y = y) = \mathbb{E}_{\hat{X} \sim \pi(X|Y=y)}\left[f_{\mathcal{X}}(\hat{X})\right]$$

$$= \mathbb{E}_{\hat{X} \sim \pi(X|Y=y)}\pi(C = 1|\hat{X})$$

$$= \int_{\mathcal{X}} \pi(C = 1|\hat{X} = \hat{x})\pi(\hat{X} = \hat{x}|Y = y)d\hat{x}$$

$$= \int_{\mathcal{X}} \pi(C = 1|\hat{X} = \hat{x}, Y = y)\pi(\hat{X} = \hat{x}|Y = y)d\hat{x}$$

$$= \pi(C = 1|Y = y),$$

where the second-to-last transition holds due to the Markov chain assumption. Thus, we obtain $f_{\mathcal{Y}}(Y) = \pi(C = 1|Y)$, completing the proof.

## A.2  OPTIMALITY FOR GENERAL CLASSIFIERS

Here, we show that the ESC (Definition 3.2), denoted as $f_{\mathcal{Y}}(Y)$, is an optimal estimator of $f_{\mathcal{X}}(X)$, regardless of the optimality of $f_{\mathcal{X}}$.

**Theorem.** *Let $X$ be the source ECG signal whose observation is the PPG signal $Y$. Additionally, assume the conversion model $g$ is a posterior sampler such that $g(Y, Z) = \hat{X} \sim \pi(X|Y)$ and let $f_{\mathcal{Y}}(Y)$ denote the ESC, as defined by Definition 3.2. Then, $f_{\mathcal{Y}}(Y)$ constitutes the minimum mean square error (MMSE) estimator of $f_{\mathcal{X}}(X)$. Namely,*

$$f_{\mathcal{Y}}(Y) = \arg\min_{\hat{f}(Y)} \mathbb{E}\left[\left(f_{\mathcal{X}}(X) - \hat{f}(Y)\right)^2 \Big| Y\right]$$

*where the expectation above is taken over $X$ given $Y$.*

The proof follows a straightforward approach. We first expand the squared error term

$$\mathbb{E}\left[\left(f_{\mathcal{X}}(X) - \hat{f}(Y)\right)^2 \middle| Y\right] = \mathbb{E}[f_{\mathcal{X}}(X)^2|Y] - 2\hat{f}(Y)\mathbb{E}[f_{\mathcal{X}}(X)|Y] + \hat{f}(Y)^2.$$

Differentiating the above expression with respect to $\hat{f}(Y)$ and setting it to zero yields

$$\hat{f}^*(Y) = \mathbb{E}[f_{\mathcal{X}}(X)|Y].$$

Recognizing that $f_{\mathcal{Y}}(Y) = \mathbb{E}_{\hat{X} \sim \pi(X|Y)}\left[f_{\mathcal{X}}(\hat{X})\right] = \mathbb{E}[f_{\mathcal{X}}(X)|Y]$, we conclude that

$$f_{\mathcal{Y}}(Y) = \hat{f}^*(Y).$$

Therefore, $f_{\mathcal{Y}}(Y)$ is the MMSE estimator of $f_{\mathcal{X}}(X)$.

## B  ASSESSING RELIABLE PPG SIGNALS VIA CALIBRATION

Recalling the desired selective guarantee specified in Equation (2),

$$\mathbb{P}\left(\frac{\mathbb{E}_{\pi(Y,C)}\left[\mathbf{1}\{\hat{C} \neq C\}\mathbf{1}\{\kappa(Y) > \lambda\}\right]}{\mathbb{E}_{\pi(Y,C)}\left[\mathbf{1}\{\kappa(Y) > \lambda\}\right]} < \alpha\right) > 1 - \delta.$$

To ensure the selection of valid PPG signals meeting the above guarantee, a calibration scheme is necessary. This scheme's goal is to identify a calibrated parameter $\hat{\lambda} \in \mathbb{R}^+$ that maximizes coverage of selected PPG signals whose classification error is below a user-specified bound, denoted by $\alpha \in (0,1)$. Then, a new unseen PPG signal, $Y \sim \pi(Y)$, will be considered reliable if its confidence score $\kappa(Y)$ surpasses $\hat{\lambda}$.

The calibration algorithm is based on a *conformal prediction* scheme (Vovk et al., 2005; Papadopoulos et al., 2002; Lei & Wasserman, 2014), a general method for distribution-free risk control. Specifically, we employ the *Learn Then Test* (LTT) (Angelopoulos et al., 2021) procedure in this paper. The conformal prediction's objective is to bound a user-defined risk function, denoted as $R(\lambda) \in (0,1)$, by adjusting a calibration parameter $\lambda \in \mathbb{R}^+$ based on empirical data, aiming for the tightest upper bound feasible (Angelopoulos & Bates, 2021). Through this calibration process, our aim is to identify $\hat{\lambda}$ that ensures the following guarantee:

$$\mathbb{P}(R(\hat{\lambda}) < \alpha) > 1 - \delta, \tag{6}$$

where $\alpha \in (0,1)$ is an upper bound for the risk function, and $\delta \in (0,1)$ is the calibration error. These parameters are user-defined and should approach zero for an effective calibration.

We now turn to describe the steps for calibration to achieve the guarantee outlined in Equation (2). To simplify Equation (2) into the form posed in Equation (6), we define the *selective risk* as:

$$R(\lambda) := \frac{\mathbb{E}_{\pi(Y,C)}\left[\mathbf{1}\{\hat{C} \neq C\}\mathbf{1}\{\kappa(Y) > \lambda\}\right]}{\mathbb{E}_{\pi(Y,C)}\left[\mathbf{1}\{\kappa(Y) > \lambda\}\right]}. \tag{7}$$

Intuitively, this measure quantifies the classification error among selected PPG signals with confidence scores surpassing $\lambda$.

In practical scenarios, computing the above risk function, $R(\lambda)$, is not feasible due to the unavailability of the joint distribution $\pi(Y,C)$. However, we do have samples from this distribution within our calibration data, $\{(Y_i, C_i)\}_{i=1}^m$. Hence, we utilize the *empirical selective risk*, denoted as $\hat{R}(\lambda)$, and defined as:

$$\hat{R}(\lambda) := \frac{\sum_{i=1}^m \mathbf{1}\{\hat{C}_i \neq C_i\}\mathbf{1}\{\kappa(Y_i) > \lambda\}}{\sum_{i=1}^m \mathbf{1}\{\kappa(Y_i) > \lambda\}}. \tag{8}$$

Note that $\hat{C}_i = \mathbf{1}\{f_{\mathcal{Y}}(Y_i) > t\}$, where $t \in \mathbb{R}^+$ represents the decision threshold.

For each $\lambda \in \Lambda$, where $\Lambda$ represents a set of calibration parameter values defined by the user as $\Lambda := [0 \ldots \lambda_{\max}]$, we utilize the empirical risk function mentioned earlier to establish an *upper-confidence bound*, denoted as $\hat{R}^+(\lambda)$, and defined as:

$$\hat{R}^+(\lambda) := \hat{R}(\lambda) + r_\delta \, , \tag{9}$$

where $r_\delta \in (0, 1)$ serves as a concentration bound, such as Hoeffding's bound, which is calculated as $r_\delta = \sqrt{\frac{1}{2m} \log \frac{1}{\delta}}$. It is shown in Hoeffding (1994) that $\mathbb{P}(\hat{R}^+(\lambda) < R(\lambda)) < \delta$.

Finally, to achieve the desired guarantee outlined in Equation (2), we seek calibration parameters that satisfy Equation (2) while using the upper confidence bound defined in Equation (9). The goal is to maximize the coverage of selected PPG signals, thus the smallest $\lambda \in \Lambda$ that meets Equation (2) is selected as follows:

$$\hat{\lambda} := \min\{\lambda : \hat{R}^+(\lambda) < \alpha, \lambda \in \Lambda\} \, . \tag{10}$$

The algorithm below summarizes the steps outlined in this section for applying calibration.

---

**Algorithm 2** Calibration of a Selective Classification UA-P2E Framework

---

**Require:** Calibration set $\{(Y_i, C_i)\}_{i=1}^m$. Selective classification error level $\alpha \in (0, 1)$. Calibration error level $\delta \in (0, 1)$. Pretrained conditional stochastic generative model $g : \mathcal{Y} \times \mathcal{Z} \to \mathcal{X}$. Pretrained classification model $f_\mathcal{X} : \mathcal{X} \to \mathbb{R}^+$. Number of samples $K > 0$. Decision threshold $t > 0$.
1: **for** $i = 1$ to $m$ **do**
2:     $\hat{C}_i, f_\mathcal{Y}(Y_i) \leftarrow$ Apply Algorithm 1 using $Y_i$, $g$, $f_\mathcal{X}$, $K$, and $t$.
3:     $\kappa(Y_i) \leftarrow \max\{1 - f_\mathcal{Y}(Y_i), f_\mathcal{Y}(Y_i)\}$
4: **end for**
5: **for** $\lambda \in \Lambda$ **do**
6:     $\hat{R}^+(\lambda) \leftarrow \frac{\sum_{i=1}^m \mathbf{1}\{\hat{C}_i \neq C_i\} \mathbf{1}\{\kappa(Y_i) > \lambda\}}{\sum_{i=1}^m \mathbf{1}\{\kappa(Y_i) > \lambda\}} + \sqrt{\frac{1}{2m} \log \frac{1}{\delta}}$
7: **end for**
8: $S := \{\lambda : \hat{R}^+(\lambda) < \alpha, \lambda \in \Lambda\}$
9: **if** $S \neq \phi$ **then** $\hat{\lambda} \leftarrow \min S$ **else** Calibration failed.
**Ensure:** Given a new PPG signal $Y \sim \pi(Y)$, **if** $\kappa(Y) > \hat{\lambda}$ **then** $Y$ is reliable.

---

## C    CONVERSION-CLASSIFICATION UNCERTAINTY EVIDENCE

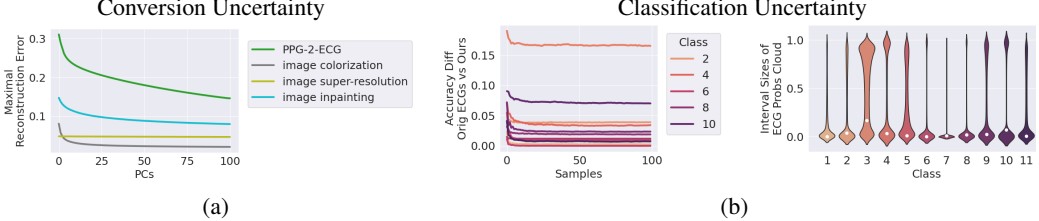

(a)                                     (b)

Figure 5: **CinC Results**: Empirical evidence of inherent uncertainty in the PPG-2-ECG conversion-classification framework. (a) The conversion spread and variability as proposed by Belhasin et al. (2023). High reconstruction error with many PCs indicates high uncertainty in terms of spread and variability. (b) Classification uncertainty shows relatively small interval sizes of ECG probability scores for each PPG (right), which can be effectively modeled using just 100 ECG samples (left).

Here, we present empirical evidence of the inherent uncertainty in the proposed PPG-2-ECG conversion, along with the classification uncertainty implied. Figure 5 summarizes the results that quantify these uncertainty layers, referring to the CinC (Reyna et al., 2021) dataset.

In Figure 5a, we illustrate the uncertainty as proposed by Belhasin et al. (2023), showing the maximal reconstruction error of the projected ground truth ECG signals among $90\%$ of the temporal signal values using a varying number of *Principal Components* (PCs) of the ECG solutions' cloud. Intuitively, a high reconstruction error using many PCs indicates high uncertainty in terms of the

spread and variability of possible solutions to the inverse problem tackled. For additional information and details on this metric, we refer the reader to previous work by Belhasin et al. (2023).

Figure 5a compares the reconstruction error for our PPG-2-ECG conversion with three similar graphs that correspond to image restoration tasks (colorization, super-resolution and inpainting). This comparison exposes the tendency of our task to have a much wider uncertainty, as far more than 100 PCs are required in order to reconstruct the ground truth ECG signals with a small error. This exposes the fact that PPG-2-ECG conversion is highly ill-posed. Put bluntly, this means that given a PPG signal, the ECG corresponding to it could be one of a wide variety of possibilities. This fact undermines our task, as it suggests that too much information has been lost in migrating from ECG to PPG, to enable a "safe" come-back. However, we argue that this is a misleading observation, and, in fact, the true uncertainty is much better behaved. More specifically, within the variability that our posterior sampler exposes, many changes manifested in the obtained ECG signals are simply meaningless and medically transparent. Indeed, if we migrate from the signal domain to classification, this uncertainty shrinks to a meaningful and crisp decisions.

The above takes us to the results shown in Figure 5b, depicting the uncertainty in classification, as arises from potential variability in the conversion process. Inspired by Angelopoulos et al. (2022), in Figure 5b (right) we present histograms of interval sizes that contain 90% of the probability scores extracted using the ECG cloud that corresponds to each PPG signal, denoted as $\{f_{\mathcal{X}}(\hat{X}_i)\}_{i=1}^{K}$. These histograms refer to the series of 11 cardiovascular conditions. Small intervals in this context correspond to a high certainty in classification, and the results reveal that for most of the conditions assessed this is indeed the case. We also see that a small fraction of PPG signals may exhibit a wide range of possible probabilities among ECG candidates, a fact that may motivate our later selective classification approach.

Figure 5b (left) presents the classification accuracy gaps for the 11 cardiac conditions mentioned. These gaps correspond to the difference between the accuracy of original ECG classification, which serves as an upper bound for our performance, and the accuracy of the average of ECG classification scores, as summarized in Algorithm 1. As can be seen, these graphs show a rapid stabilization, suggesting that using 100 samples is definitely sufficient for modeling the uncertainty that arises from the conversion process in the classification domain.

## D  ANALYSIS OF OVERFITTING IN THE CONVERSION MODEL

In this section, we provide additional experimental evidence supporting the validity of our conversion model using the MIMIC-III (Johnson et al., 2016) dataset. We analyze the effect of overfitting and demonstrate that our model does not exhibit overfitting.

### D.1  CONVERSION TRAINING METRICS: TRAINING VS. VALIDATION

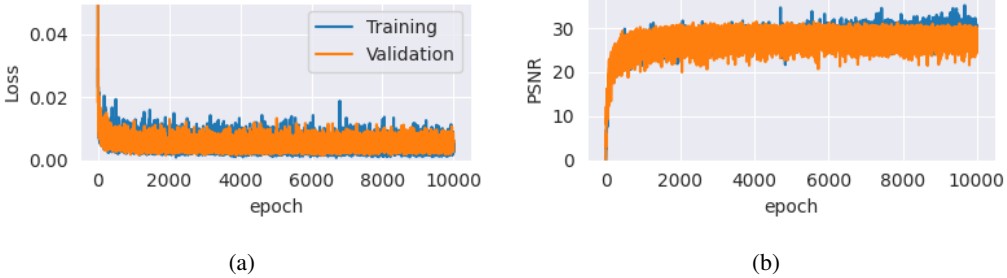

(a)                                            (b)

Figure 6: **MIMIC-III Results**: Metrics report of MSE loss and PSNR comparing the training set and validation set during training, indicating no overfitting.

Our PPG-2-ECG conversion model is a conditional diffusion model trained on the MIMIC-III (Johnson et al., 2016) dataset. Following the gold standards for training conditional diffusion models, a denoising model is required to clean noise from noisy ECG signals, conditioned on the PPG input.

In Figure 6, we present the average MSE loss and PSNR of the denoised ECG signals compared to the original ones for each epoch during the training process. It is evident that the performance on the training and validation sets is similar, indicating that there is no overfitting. We attribute this to the diversity and large scale of the MIMIC-III dataset.

## D.2 ORIGINAL ECG FREQUENCY APPROXIMATION

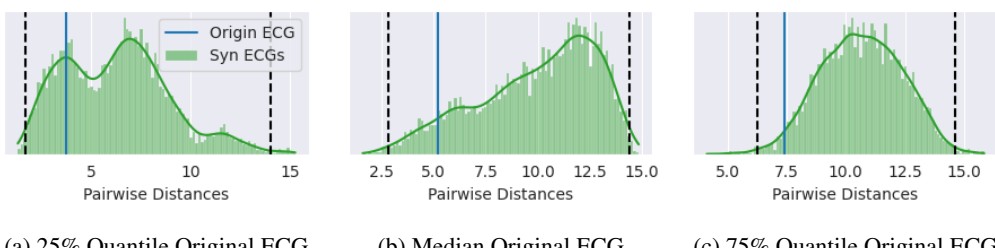

    (a) 25% Quantile Original ECG      (b) Median Original ECG      (c) 75% Quantile Original ECG

Figure 7: **MIMIC-III Results**: Pairwise synthetic ECG distances (GREEN) for three PPG signals and the nearest synthetic ECG distance to the original ECG (BLUE). Results show the original ECG's approximation within the synthetic solution cloud across three distance quantiles of each original ECG.

Here, we present an additional signal-quality assessment, demonstrating how often the original ECG can be approximated as one of the synthetic ECG signals within each solution cloud for each PPG.

To evaluate this, we first calculated histograms of pairwise euclidean distances between synthetic ECG signals within the solution cloud for each PPG signal in the test set. Next, we determined the distance from the original ECG to its nearest synthetic ECG signal. Then, we assessed how often the distance of the original ECG fell within the 99% of the histogram. We found that **93.23%** of the original ECG signals were contained within the solution clouds across the test set, indicating that our solution cloud provides a suited approximation of the original ECG signals.

In Figure 7, we visually demonstrate the inclusion of the original ECG distance within three histograms of three different PPG signals.

# E  EXPERIMENTAL DETAILS

This section provides a thorough description of the experimental methodology utilized in our paper. It encompasses the datasets employed for both training and evaluating the conversion and classification models, along with details regarding data pre-processing and training schemes.

## E.1  DATASETS AND PREPROCESSING

In this work we rely on two datasets: the MIMIC-III matched waveform database (Johnson et al., 2016), which consists of pairs of measured ECG signals, denoted as $X$, and corresponding PPG signals, denoted as $Y$; and the "Computing in Cardiology" dataset (CinC) (Reyna et al., 2021), which contains pairs of measured ECG signals along with associated cardiovascular condition labels, referred to as $C$.

### E.1.1  MIMIC-III DATASET SUMMARY

MIMIC-III is one of the most extensive datasets for paired PPG and ECG signals. It comprises 22,317 records from 10,282 distinct patients, with each record spanning approximately 45 hours.

The MIMIC-III dataset includes a substantial portion of noisy signals, which are unsuitable for training models due to various distortions, artifacts, synchronization issues, and other anomalies. Consequently, a comprehensive preprocessing protocol is employed for preparing the data for subsequent analysis. The preprocessing of each record in MIMIC-III is conducted through the following steps:

1. **Resampling**: Each record is resampled to a uniform sampling rate of 125 Hz.

2. **Signal Smoothing**: A zero-phase 3rd order Butterworth bandpass filter with a frequency band ranging from 1 Hz to 47 Hz is applied to all signals to reduce noise.

3. **Signal Alignment**: Ensures that all signals are temporally aligned.

4. **Artifact Removal**: Signal blocks with unacceptable heart rates, as extracted from the electrocardiogram (ECG), are removed.

5. **Consistency Check**: Removes signal blocks where the heart rates derived from the photoplethysmogram (PPG) differ by more than 30% from those extracted from the ECG.

6. **Trend Removal**: The global trend (DC-drift) in the signal is removed to prevent bias.

7. **Normalization**: All signals are normalized to have zero mean and unit variance.

### E.1.2 CINC DATASET SUMMARY

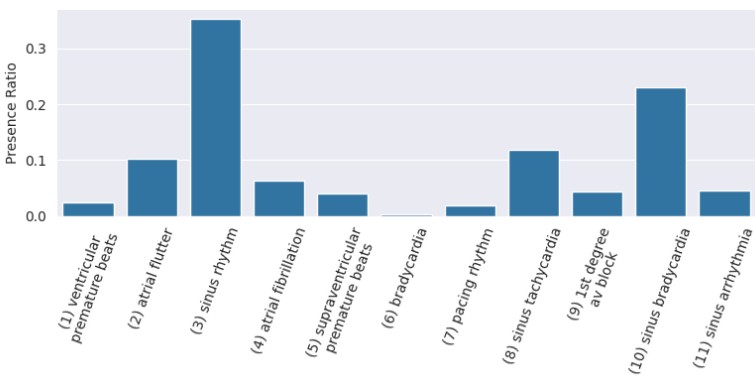

Figure 8: **CinC Results**: Class distribution report, indicating a significant class imbalance.

Moving to CinC, which to the best of our knowledge, is the largest and most challenging classification dataset in cardiology, containing 88,253 of 10-second multi-labeled ECG recordings from distinct patients. CinC integrates data from several public databases, including the CPSC Database and CPSC-Extra Database, INCART Database, PTB and PTB-XL Database, the Georgia 12-lead ECG Challenge (G12EC) Database, Augmented Undisclosed Database, Chapman-Shaoxing and Ningbo Database, and the University of Michigan (UMich) Database. For further details on these databases, we refer the reader to Reyna et al. (2021).

Each recording is labeled with the presence or absence (binary label) of various cardiovascular conditions. From these, 11 conditions detectable using lead II ECG alone were selected: (1) Ventricular premature beats, (2) Atrial flutter, (3) Sinus rhythm, (4) Atrial fibrillation, (5) Supraventricular premature beats, (6) Bradycardia, (7) Pacing rhythm, (8) Sinus tachycardia, (9) 1st degree AV block, (10) Sinus bradycardia, and (11) Sinus arrhythmia. We note that the labels considered in CinC exhibit significant imbalance, as presented in Figure 8.

The CinC database, noted for its cleaner and more structured data, lacks the pairings and complexities found in the MIMIC-III dataset and only includes ECG samples. For this dataset, the preprocessing steps are slightly modified and include:

1. **Resampling**: Each record is resampled to 125 Hz to standardize the data input.

2. **Signal Smoothing**: A zero-phase 3rd order Butterworth bandpass filter with a frequency band ranging from 1 Hz to 47 Hz is employed to smooth the ECG signals.

3. **Trend Removal**: Global trends are removed.

4. **Normalization**: Signals are normalized to have zero mean and unit variance.

E.2 TRAINING DETAILS FOR THE CONVERSION MODEL

Our study leverages a diffusion-based conversion methodology. Specifically, we adopt the DDIM framework (Song et al., 2020). All the training details for the reversed conversion model, designed for generating PPG signals given ECG ones, are similar to the training details for the ECG-2-PPG conversion, as described hereafter.

The denoising model employed in our experiments is based on a U-Net (Ronneberger et al., 2015) architecture with an attention mechanism, as proposed in Ho et al. (2020); Dhariwal & Nichol (2021). Originally introduced in the field of image processing, this architecture comprises a series of residual layers and downsampling convolutions, followed by a subsequent series of residual layers with upsampling convolutions. Additionally, a global attention layer with a single head is incorporated, along with a projection of the timestep embedding into each residual block. To adapt the architecture for our purposes, we replace the convolutions and attention layers with 1D counterparts, enabling training and evaluation directly in the raw 1D signal domain. In our study we use $d = 1024$ PPG and ECG signal length.

Regarding hyperparameters for the U-Net architecture, we set 256 dimensions for time embedding. The downsampling and upsampling is applied by factors of [1, 1, 2, 2, 4, 4, 8], with each factor corresponding to a stack of 2 residual blocks. Attention is applied after the downsampling/upsampling matching to the factor of 4. To address overfitting, we incorporate dropout with a ratio of 0.2.

Concerning training, the model architecture is trained on MIMIC-III. We split the data into training and validation sets after preprocessing. Specifically, we randomly select 400 patients for the validation set while the remaining patients use for training. For each patient in the training set we split the raw PPG/ECG signals into pairs of 10-seconds (length 1024) signals, resulting in 15,225,389 PPG/ECG signals. For each patient in the validation set, we sample 100 distinct 10-seconds (length 1024) PPG/ECG signals from the raw signals, resulting in 40,000 PPG/ECG signals.

The model architecture is trained for 10,000 epochs, employing a batch size of 2,048 PPG and ECG recordings across 8×V100 GPUs. The noise schedule progresses linearly over 1,000 time steps, starting at 1e-6 and ending at 1e-2. For optimization, we employed the standard Adam optimizer with a fixed learning rate of 1e-4. An $L_2$ loss is utilized to estimate the noise in the input instance.

For evaluation, we sample $K = 100$ ECG candidates for each PPG signal using the diffusion-based framework, and repeat the generation process for 3 different seeds. We adopt a non-stochastic diffusion process as proposed in Song et al. (2020), with skip jumps of 10 iterations, resulting in a total of $T = 1000/10 = 100$ diffusion iterations.

E.3 TRAINING DETAILS FOR THE CLASSIFICATION MODEL

Our paper adopts a classification methodology over the CinC dataset, referred to as $f_{\mathcal{X}}$, which classifies ECG signals, whether original or synthesized, into the 11 cardiovascular conditions in CinC. Additionally, we report the classification performance of a direct approach, denoted as $f_{\mathcal{Y}}$, which directly gets PPG signals instead of ECG ones. The training scheme outlined hereafter applies to both classifiers.

As discussed earlier, the presence of each cardiovascular condition in the CinC dataset is imbalanced, and thus CinC poses significant challenges for developing a robust classifier. In Figure 9 (red) we report the frequency of each cardiovascular condition in the dataset. Note that the challenge in sampling class-balanced batches arises from the fact that samples may be positive for multiple cardiovascular conditions at once. To address this issue, we implement a novel class-balance sampling technique. Given the multi-label nature of the classification, where class dependencies exist, our strategy not only ensures a uniform class distribution for positive samples but also carefully selects negative samples from a balanced pool reflecting common class co-occurrences. This approach significantly aids in achieving a more uniform label distribution, the effectiveness of which is demonstrated in Figure 9 (blue).

Specifically, we consider a training set $S_{\text{Train}}$, where each pair $(X, C) \in S_{\text{Train}}$ satisfies $X \in \mathcal{X}$ and $C \in \{0, 1\}^{L_{\max}}$. Here, $C$ is a binary vector where $C_i = 1$ if and only if the label $i$ is present in $X$. At each timestep, we first sample a random label uniformly from the $L_{\max} = 11$ cardiovascular conditions. We then ensure that both positive $(X^p, C^p)$ and negative $(X^n, C^n)$ samples for this

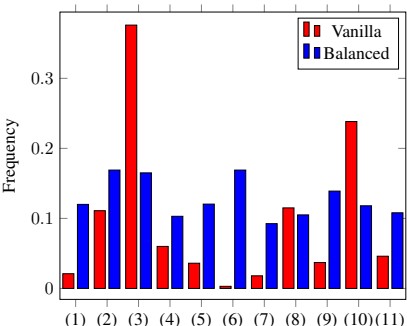

Figure 9: An illustration of the label histograms for two different sampling methods on the CinC training dataset: vanilla-sampling and balanced-sampling. In the histogram for vanilla sampling, the distribution of labels is as they naturally appear in the dataset. On the other hand, in the histogram for balanced sampling, the labels are intentionally adjusted to achieve a more uniform distribution.

condition are included in the batch. Additionally, we control the ratio of the presence of the major label, denoted by $1 \le l \le L_{\max}$, as one condition is often predominant, representing the "normal" condition. In our case, this label is sinus rhythm (see Figure 9). We set this ratio to be $p_l = 0.2$. For a formal summary of this method, please refer to Algorithm 3.

---

**Algorithm 3** Class-Balanced Batch Sampling

---

**Require:** Training set $S_{\text{Train}} \subseteq \mathcal{X} \times \mathcal{C}$ s.t. $\mathcal{C} = \{0,1\}^{L_{\max}}$. Batch size $2b \in \mathbb{N}$. Major label index $1 \le l \le L_{\max}$.
    Major label ratio $p_l \in [0,1]$.
1: $S_{\text{Batch}} \leftarrow \{\}$
2: **for** $i = 1$ to $b$ **do**
3:     $L \sim U(\{1, 2, \ldots, L_{\max}\})$
4:     $(X^p, C^p) \sim U(\{(X,C) \in S_{\text{Train}} : C_L = 1\})$
5:     **if** $L = l$ **then** $(X^n, C^n) \sim U(\{(X,C) \in S_{\text{Train}} : C_L = 0\})$
6:     **else** $(X^n, C^n) \sim U(\{(X,C) \in S_{\text{Train}} : C_L = 0, C_l \sim B(p_l)\})$
7:     $S_{\text{Batch}} \leftarrow \{(X^p, C^p), (X^n, C^n)\}$
8: **end for**
**Ensure:** Class-balanced batch $S_{\text{Batch}}$.

---

The classification architecture utilized in our studies is based on a simplified VGG (Simonyan & Zisserman, 2014) model with 8 layers, incorporating batch normalization modules to promote stability, along with dropout at ratio of 0.2 to address overfitting. Similar to the conversion model, we modify the VGG architecture for 1D input signals by reconfiguring the convolutional layers for 1D operation.

The training process is implemented above CinC using a cross-validation scheme with 3 seeds, where for each seed, we create a separated 80-20 training and validation set patient split after pre-processing. Specifically, our training sets contain 65,535 10-seconds (length 1024) ECG signals, while the validation sets contain 16,386 10-seconds (length 1024) ECG signals.

Optimization is performed on $1 \times$V100 GPU using the Adam optimizer and an $L_2$ weight decay of 1e-4. The model is trained over 20 epochs with a batch size of 128, employing a learning rate of 1e-3 with a linear decay and using binary cross-entropy loss as the loss function. Since the primary focus of our paper is on the conversion task rather than the classification tasks, which are used for validation and to motivate the consideration of uncertainty in the conversion, the decision thresholds were simply fixed to 0.5 as standard.

## F  CLASSIFICATION STRATEGIES

In our experiments we assess several classification strategies, all relying on paired ECG and PPG signals and their corresponding labels, as obtained using MIMIC-III (Johnson et al., 2016) and CinC (Reyna et al., 2021). We assume the availability of pretrained classifiers, $f_{\mathcal{X}} : \mathcal{X} \to \mathbb{R}^+$ and

$f_{\mathcal{Y}} : \mathcal{Y} \to \mathbb{R}^+$, for input ECG and PPG signals, respectively. These classifiers output classification likelihoods considering cardiovascular conditions.

We emphasize that the baseline approaches, as utilized in prior work, ignore the uncertainty in the conversion of PPG to ECG by maintaining only a single ECG solution given the PPG, whereas our approach maintains and manipulates an aggregation of multiple ECG candidates for the classification. With this in mind, here are the classification strategies considered in this work:

**Original ECG:** Classifier performance when tested on original ECG signals, i.e., the performance of $f_{\mathcal{X}}(X)$, where $X \sim \pi(X)$. Since we do not have access to the original ECG signal in our PPG-2-ECG setting, this measure serves as an upper bound for the achievable performance.

**Original PPG:** Classifier performance when tested on the original PPG signals, i.e., the performance of $f_{\mathcal{Y}}(Y)$, where $Y \sim \pi(Y)$.

**Synthesized PPG:** Classifier performance when tested on diffusion-based random PPG samples, i.e., the performance of $f_{\mathcal{Y}}(\hat{Y})$, where $\hat{Y} \sim \hat{\pi}(Y|X)$ are outcomes of the reversed diffusion-based conversion model. See Section 5.2 for detailed explanation on the need for these signals.

**Synthesized ECG: SSC Mean** Classifier performance when tested on single-solution (Definition 3.1) ECG signals, as practiced by prior work, referring to the mean ECG signal from the diffusion-based ECG samples, i.e., the performance of $f_{\mathcal{X}}(\frac{1}{K}\sum_{i=1}^{K}\hat{X}_i)$, where $\hat{X}_i \sim \hat{\pi}(X|Y)$.

**Synthesized ECG: SSC Random:** Classifier performance when tested on single-solution (Definition 3.1) ECG signals, as practiced by prior work, referring to randomly chosen diffusion-based ECG samples, i.e., the performance of $f_{\mathcal{X}}(\hat{X})$, where $\hat{X} \sim \hat{\pi}(X|Y)$.

**Synthesized ECG: ESC (Ours):** Classifier performance when tested on the proposed multi-solution (Definition 3.2 and Algorithm 1) ECG signals referring to randomly chosen diffusion-based ECG samples, and then averaging their classification scores, i.e., the performance of $\frac{1}{K}\sum_{i=1}^{K}f_{\mathcal{X}}(\hat{X}_i)$, where $\hat{X}_i \sim \hat{\pi}(X|Y)$.

Note that neither RDDM (Shome et al., 2023) nor CardioGAN (Sarkar & Etemad, 2021) provides publicly available code, which are currently state-of-the-art conversion models. This limits our ability to generate ECG signals using their methods and evaluate their classification performance. However, our evaluation setup is more comprehensive, making CardioGAN and RDDM comparable to the evaluated classification strategies. Both approaches combine the MMSE estimate with a random single sample, since they both rely on a single random ECG solution that tends toward the mean due to mode collapse, as shown in Table 1.

## G    EXPLORING THE VALIDITY OF SYNTHESIZED PPG SIGNALS

The main question arising from the results depicted in Section 5.2 is whether using synthesized PPG signals for drawing the various conclusions is a fair and trust-worthy strategy. One potential medical rationale for this approach is that PPG signals inherently encompass noise. Consequently, employing synthesized PPG signals, even if noisy themselves, can accurately model real-world scenarios. Still, in order to address this question in a more direct form, we return to the MIMIC-III (Johnson et al., 2016) dataset, which contains true PPG signals, and conduct three additional studies described hereafter.

### G.1    ACHIEVING SIMILAR TRENDS WITH TRUE PPG SIGNALS AND TRUE LABELS

First, we use a much smaller held-out subset of the MIMIC-III database that contains binary labels indicating the presence of atrial fibrillation (AFib). Therefore, for this heart condition, we can evaluate classification performance for both true PPG signals and synthesized ones.

As previously stated in Section 5.2, the MIMIC-III matched waveform database lacks cardiovascular labels. However, textual cardiac reports are available for a subset of patients within MIMIC-III, referred to as the MIMIC-III clinical database (Johnson et al., 2016). In order to extract AFib labels from this subset, we filtered 400 patients, with 200 reported cases of AFib and 200 with normal sinus rhythm. The determination of AFib or sinus rhythm presence was made by searching for the

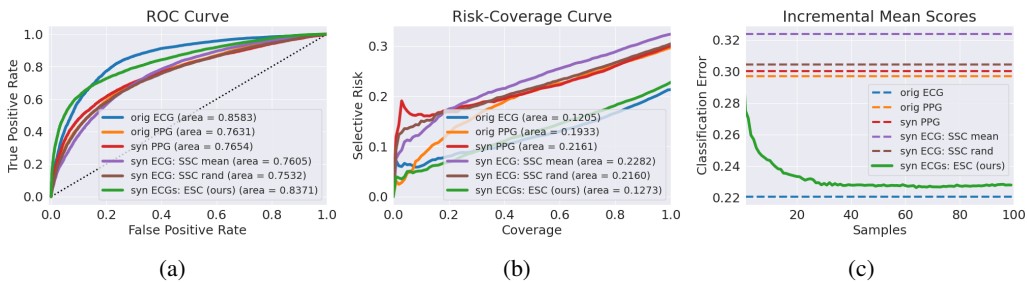

Figure 10: **MIMIC-III Results**: (a) ROC Curves illustrating classification performance. (b) Risk-Coverage curves (Geifman & El-Yaniv, 2017) demonstrating uncertainty quantification performance. (c) Incremental classification error as a function of the number of ECG solutions utilized for each PPG signal.

terms "atrial fibrillation" and "sinus rhythm" within the cardiac reports. We note that extracting labels using this searching approach may provide erroneous labels.

In Figure 10 we bring the classification results in an experiment that follows the ones performed on CinC, with one distinct difference – we assess both real and synthesized PPG signals. Broadly speaking, all the conclusions from Section 5.2 generally hold, namely (i) The best performance is obtained by classifying ECG directly; (ii) Our approach is superior to all other alternatives, including a direct classification of PPG (be it real or synthesized).

Focusing on the main theme of this experiment, we see that the performance of classifying the original PPG (ORANGE) is comparable to the alternative of operating on the synthetic signals (RED). In terms of AUROC, the performances are nearly identical. When evaluating the area under the Risk-Coverage curve (Geifman & El-Yaniv, 2017), we observe a similar trend; the uncertainty performance of both approaches is almost identical when considering around half of the test samples. These findings offer empirical evidence supporting the validity of our reversal conversion model.

### G.2 ASSESSING THE HIGH QUALITY OF SYNTHETIC PPG SIGNALS

| Syn PPG | 1-FD ↓ | 100-FD ↓ |
|---|---|---|
| DDIM (T=25) | $3.8365 \pm 0.0021$ | $3.8248 \pm 0.0013$ |
| DDIM (T=50) | $0.4049 \pm 0.0013$ | $0.3927 \pm 0.0007$ |
| DDIM (T=100) | $\mathbf{0.1181 \pm 0.0011}$ | $\mathbf{0.1036 \pm 0.0002}$ |

Table 3: **MIMIC-III Results**: Quality assessment of our reversed diffusion-based conversion model. The results show the mean metric estimates and their standard errors across the three seeds.

Second, we conduct an experiment that evaluates the signal quality of the reversed conversion model over MIMIC-III. Specifically, given the ECG-2-PPG algorithm, we generate many PPG signals; we then compute the distance between the distributions of the true PPG and the synthesized ones. The results are presented in Table 3, which is similar to Table 1, showing (with T=100) that the distributions are nearly indistinguishable from each other (with almost zero FD). Furthermore, the quality of the synthesized PPG signals improves with more diffusion iterations.

### G.3 ASSESSING THE HIGH QUALITY OF SYNTHETIC ECG SIGNALS FROM SYNTHETIC PPG ONES

Finally, we conduct an experiment that testifies the signal quality after cycle transformation: ECG to PPG to ECG. Specifically, we sample (synthetic) PPG signals from true ECG signals in MIMIC-III, then re-sample (synthetic) ECG signals from these synthetic PPG ones. Finally, we measure their distribution distance compared to the true ECG signals. The results are presented in Table 4, which is also similar to Table 1. These results indicate that the synthetic ECG signals derived from synthetic PPG are almost as high-quality as those derived from true PPG. The negligible loss in

| UA-P2E | 1-FD ↓ | 100-FD ↓ |
|---|---|---|
| Syn ECGs from True PPG | $0.3198 \pm 0.0020$ | $0.2379 \pm 0.0005$ |
| Syn ECGs from Syn PPG | $0.3940 \pm 0.0023$ | $0.3164 \pm 0.0005$ |

Table 4: **MIMIC-III Results**: Quality assessment of our synthetic ECG signals derived from synthetic PPG vs those derived from true PPG. The results show the mean metric estimates and their standard errors across the three seeds.

quality is expected, given that the synthetic PPG signals are not perfect. This empirical evidence further supports the use of synthetic PPG data in our work.

# H    SUPPLEMENTARY EXPERIMENTAL ANALYSIS OF CINC RESULTS

This section extends the empirical classification results referring to CinC (Reyna et al., 2021), that were outlined in Section 5.2.

## H.1    REPORTED NUMERICAL RESULTS

| Classification Strategy | Macro-AUROC ↑ | Macro-AURC ↓ |
|---|---|---|
| Original ECG | $0.9435 \pm 0.0020$ | $0.0048 \pm 0.0000$ |
| Synthesized PPG | $0.7857 \pm 0.0021$ | $0.0313 \pm 0.0001$ |
| Synthesized ECG: SSC Mean | $0.7286 \pm 0.0068$ | $0.0405 \pm 0.0017$ |
| Synthesized ECG: SSC Random | $0.7458 \pm 0.0007$ | $0.0392 \pm 0.0002$ |
| **Synthesized ECG: ESC (Ours)** | $\mathbf{0.8496 \pm 0.0010}$ | $\mathbf{0.0211 \pm 0.0001}$ |

Table 5: **CinC Results**: Mean and standard error values of macro-level AUROC and AURC across three random seeds. See Appendix F for the classification strategies considered.

Here, we provide the numerical results of the mean metric values along with their corresponding standard errors reported in Figure 3 and Figure 4.

Tables 5 and 7 display the AUROC (Area Under the ROC curve) and AURC (Area Under the Risk-Coverage curve (Geifman & El-Yaniv, 2017)) metrics. Specifically, Table 7 outlines AUROC/AURC mean and standard error values associated with each classification strategy per cardiovascular condition, whereas Table 5 provides macro-level metrics derived by averaging over all examined cardiovascular conditions.

## H.2    EVALUATING ADDITIONAL METRICS

| Classification Strategy | Macro-TPR ↑ | Macro-TNR ↑ | Macro-F1 ↑ |
|---|---|---|---|
| Original ECG | $0.6684 \pm 0.0009$ | $0.9802 \pm 0.0002$ | $0.6679 \pm 0.0005$ |
| Synthesized PPG | $0.2850 \pm 0.0007$ | $0.9694 \pm 0.0000$ | $\mathbf{0.3139 \pm 0.0003}$ |
| Synthesized ECG: SSC Mean | $0.2542 \pm 0.0081$ | $0.9560 \pm 0.0012$ | $0.2663 \pm 0.0081$ |
| Synthesized ECG: SSC Random | $\mathbf{0.3218 \pm 0.0008}$ | $0.9509 \pm 0.0000$ | $0.3100 \pm 0.0009$ |
| **Synthesized ECG: ESC (Ours)** | $0.2902 \pm 0.0039$ | $\mathbf{0.9783 \pm 0.0002}$ | $\mathbf{0.3154 \pm 0.0017}$ |

Table 6: **CinC Results**: Mean and standard error values of macro-level sensitivity (TPR), specificity (TNR), and F1 scores, across three random seeds. See Appendix F for the classification strategies considered.

To enhance clinical relevance and facilitate comparisons with future studies, we evaluate additional metrics related to the classification performance in mitigating class imbalance. Specifically, we

evaluate sensitivity (TPR), specificity (TNR), and F1 scores across different classification strategies. In Table 6 we provide the reported results.

It is evident from Table 6 that our classification approach outperforms in specificity (TNR) compared to sensitivity (TPR). The random ECG signal classification approach, stems from the generation process of our diffusion model, demonstrates better sensitivity than multi-solution approach. We attribute this phenomenon to the rarity of various cardiovascular conditions in the CinC dataset (see Figure 8), which makes it challenging to achieve high sensitivity, even with the original ECG signals where no PPG signals are considered.

While our approach achieves superior performance in specificity, it can be used in clinical setting as a better detector for the absence of cardiovascular conditions on average, while a single ECG sample of our diffusion model is evident to be better on the detection of the presence of the cardiovascular conditions on average.

## I    WHY THE ORIGINAL ECG PERFORMANCE SETS AN UPPER BOUND FOR PPG-BASED APPROACH?

The performance of the original ECG signal represents an upper bound for classification performance, including our own method. Intuitively, this discrepancy arises because cardiovascular labels are directly linked to heart conditions determined by ECG signals, while PPG signals only capture blood volume fluctuations, which provide partial information about the heart. As a result, PPG signals are a degraded version of ECG signals, leading to a natural decline in performance due to the loss of information. Even when generating ECG candidates from PPG signals, some information loss is unavoidable, which further reduces performance. Our approach mitigates this issue by accounting for conversion uncertainty, yielding superior results compared to other PPG-based strategies.

From a more theoretical perspective, we adopt a Markovian framework: $Y \rightarrow X \rightarrow C$, where $Y$ represents the PPG signal, $X$ denotes the ECG signal, and $C$ is the cardiovascular condition label. Under this assumption, $\pi(C|X,Y) = \pi(C|X)$, meaning that, given the ECG signal $X$, the PPG signal $Y$ does not provide any additional information about the cardiovascular condition $C$. Consequently, when we degrade the class information from $X$ to $Y$, performance naturally declines due to the inherent loss of information.

| Cardiovascular Condition | Classification Strategy | AUROC ↑ | AURC ↓ |
|---|---|---|---|
| 1) Ventricular premature beat | Origin ECG | 0.8953 ± 0.0081 | 0.0042 ± 0.0005 |
| | Syn PPG | 0.6724 ± 0.0033 | 0.0216 ± 0.0009 |
| | Syn ECG: SSC Mean | 0.6290 ± 0.0194 | 0.0171 ± 0.0006 |
| | Syn ECG: SSC Random | 0.6344 ± 0.0087 | 0.0178 ± 0.0014 |
| | **Syn ECG: ESC (Ours)** | **0.8006 ± 0.0072** | **0.0077 ± 0.0006** |
| 2) Atrial flutter | Origin ECG | 0.9710 ± 0.0004 | 0.0056 ± 0.0001 |
| | Syn PPG | 0.8457 ± 0.0018 | 0.0289 ± 0.0006 |
| | Syn ECG: SSC Mean | 0.7648 ± 0.0170 | 0.0424 ± 0.0040 |
| | Syn ECG: SSC Random | 0.8018 ± 0.0035 | 0.0380 ± 0.0009 |
| | **Syn ECG: ESC (Ours)** | **0.8816 ± 0.0009** | **0.0220 ± 0.0003** |
| 3) Sinus rhythm | Origin ECG | 0.9728 ± 0.0004 | 0.0175 ± 0.0003 |
| | Syn PPG | 0.8111 ± 0.0010 | 0.1309 ± 0.0005 |
| | Syn ECG: SSC Mean | 0.7638 ± 0.0018 | 0.1700 ± 0.0012 |
| | Syn ECG: SSC Random | 0.7773 ± 0.0009 | 0.1618 ± 0.0010 |
| | **Syn ECG: ESC (Ours)** | **0.8547 ± 0.0004** | **0.1114 ± 0.0013** |
| 4) Atrial fibrillation | Origin ECG | 0.9489 ± 0.0008 | 0.0059 ± 0.0001 |
| | Syn PPG | 0.8263 ± 0.0007 | 0.0283 ± 0.0005 |
| | Syn ECG: SSC Mean | 0.8320 ± 0.0144 | 0.0262 ± 0.0004 |
| | Syn ECG: SSC Random | 0.8582 ± 0.0044 | 0.0317 ± 0.0010 |
| | **Syn ECG: ESC (Ours)** | **0.9196 ± 0.0016** | **0.0147 ± 0.0003** |
| 5) Supraventricular premature beats | Origin ECG | 0.9080 ± 0.0017 | 0.0068 ± 0.0002 |
| | Syn PPG | 0.6673 ± 0.0033 | 0.0302 ± 0.0011 |
| | Syn ECG: SSC Mean | 0.6176 ± 0.0223 | 0.0313 ± 0.0027 |
| | Syn ECG: SSC Random | 0.6404 ± 0.0054 | 0.0351 ± 0.0006 |
| | **Syn ECG: ESC (Ours)** | **0.7739 ± 0.0069** | **0.0163 ± 0.0006** |
| 6) Bradycardia | Origin ECG | 0.8072 ± 0.0224 | 0.0010 ± 0.0002 |
| | Syn PPG | **0.7734 ± 0.0154** | 0.0023 ± 0.0001 |
| | Syn ECG: SSC Mean | 0.6405 ± 0.0284 | 0.0143 ± 0.0057 |
| | Syn ECG: SSC Random | 0.6927 ± 0.0098 | 0.0028 ± 0.0005 |
| | **Syn ECG: ESC (Ours)** | **0.7853 ± 0.0113** | **0.0011 ± 0.0001** |
| 7) Pacing rhythm | Origin ECG | 0.9734 ± 0.0048 | 0.0009 ± 0.0002 |
| | Syn PPG | 0.7258 ± 0.0119 | **0.0054 ± 0.0001** |
| | Syn ECG: SSC Mean | 0.6148 ± 0.0212 | 0.0139 ± 0.0010 |
| | Syn ECG: SSC Random | 0.6193 ± 0.0181 | 0.0137 ± 0.0008 |
| | **Syn ECG: ESC (Ours)** | **0.7966 ± 0.0061** | **0.0055 ± 0.0002** |
| 8) Sinus tachycardia | Origin ECG | 0.9932 ± 0.0005 | 0.0015 ± 0.0001 |
| | Syn PPG | 0.9674 ± 0.0003 | 0.0094 ± 0.0002 |
| | Syn ECG: SSC Mean | 0.9451 ± 0.0030 | 0.0136 ± 0.0010 |
| | Syn ECG: SSC Random | 0.9263 ± 0.0030 | 0.0184 ± 0.0009 |
| | **Syn ECG: ESC (Ours)** | **0.9792 ± 0.0018** | **0.0050 ± 0.0005** |
| 9) 1st degree AV block | Origin ECG | 0.9634 ± 0.0011 | 0.0029 ± 0.0001 |
| | Syn PPG | 0.6350 ± 0.0043 | 0.0365 ± 0.0006 |
| | Syn ECG: SSC Mean | 0.6139 ± 0.0092 | 0.0332 ± 0.0009 |
| | Syn ECG: SSC Random | 0.6184 ± 0.0050 | 0.0347 ± 0.0011 |
| | **Syn ECG: ESC (Ours)** | **0.7193 ± 0.0095** | **0.0203 ± 0.0010** |
| 10) Sinus bradycardia | Origin ECG | 0.9935 ± 0.0002 | 0.0025 ± 0.0001 |
| | Syn PPG | 0.9438 ± 0.0008 | 0.0304 ± 0.0005 |
| | Syn ECG: SSC Mean | 0.8932 ± 0.0028 | 0.0565 ± 0.0028 |
| | Syn ECG: SSC Random | 0.9005 ± 0.0010 | 0.0502 ± 0.0011 |
| | **Syn ECG: ESC (Ours)** | **0.9580 ± 0.0003** | **0.0199 ± 0.0011** |
| 11) Sinus arrhythmia | Origin ECG | 0.9512 ± 0.0003 | 0.0039 ± 0.0001 |
| | Syn PPG | 0.7741 ± 0.0032 | 0.0207 ± 0.0004 |
| | Syn ECG: SSC Mean | 0.6999 ± 0.0039 | 0.0272 ± 0.0019 |
| | Syn ECG: SSC Random | 0.7344 ± 0.0037 | 0.0268 ± 0.0006 |
| | **Syn ECG: ESC (Ours)** | **0.8771 ± 0.0006** | **0.0086 ± 0.0001** |

Table 7: **CinC Results**: Mean and standard error values of AUROC and AURC (Geifman & El-Yaniv, 2017) across three random seeds, shown for each cardiovascular condition examined. See Appendix F for the classification strategies considered.

