# OpenReview forum: "Uncertainty-Aware PPG-2-ECG for Enhanced Cardiovascular Diagnosis using Diffusion Models"
_ICLR.cc/2025/Conference — Submitted to ICLR 2025_

### Official Review · Reviewer_r8zb · 2024-11-01

**Soundness:** 2
**Presentation:** 3
**Contribution:** 3
**Rating:** 5
**Confidence:** 5

**Summary:**

The paper introduces a novel methodology for synthesizing Electrocardiogram (ECG) signals from Photoplethysmogram (PPG) signals, aimed at enhancing the reliability of cardiovascular disease diagnostics while avoiding the difficulties associated with ECG acquisition. By measuring the uncertainties involved in generating the ECG from PPG signals and in classifying from the generated signals, the authors convincingly demonstrate the feasibility and superiority of their proposed method.

**Strengths:**

1. The theoretical foundation validating the transition from PPG to ECG and then to classification is robust. The paper demonstrates that generating multiple candidate ECG sample sets can mitigate the uncertainty of ECG conversion and eliminate errors due to mismatch.
2. Quantification of uncertainties during the conversion process enhances the interpretability of the results.
3. The methodological design is well-justified through practical experiments showing the reliability of PPG-derived ECGs, surpassing state-of-the-art methods. Ablation studies further substantiate the validity of the proposed approach.
4. The paper is well-written with a clear structure, making it easy to follow.

**Weaknesses:**

1. The paper should specify which particular database the CinC dataset is derived from. The rationale behind choosing these 11 types of anomalies should be clarified.
2. Although the authors provide a thorough theoretical foundation and extensive analysis, the task of generating ECG from PPG lacks inherent rationale. It is unclear whether the generated ECG can reliably reflect arrhythmias, making this approach seem like an uncertain application of analytical techniques with limited practical value.
3. The authors should further compare their classification results with state-of-the-art models, as the current performance appears suboptimal for ECG classification tasks.

**Questions:**

1. As Figure 3 shows, the performance of the PPG-derived ECG significantly trails that of the original ECG. How do you explain this discrepancy? Is it possible that the required information for detecting certain arrhythmias is inherently absent in the PPG signals?
2. Appendix G focuses on Atrial Fibrillation (AF), which is relatively easier to detect. It would be beneficial to include results for other types of diseases to provide a broader evaluation.
3. How does pacing rhythm manifest in PPG signals? Is it feasible to accurately generate ECG signals with pacing rhythm characteristics from PPG?

---

> ### Author Response · Authors · 2024-11-22
>
> W.1. “The paper should specify which particular database the CinC dataset is derived from. The rationale behind choosing these 11 types of anomalies should be clarified.”
>
> We apologize for this misinformation, and we have revised the paper to include a data summary (see Appendix E lines 1005-1010). Additionally, relevant cardiovascular conditions were selected based on their detectability via lead II, following consultation with a cardiologist. This rationale is provided in line 405 and lines 1011-1012 in the appendix.
>
> W.2. “The task… lacks inherent rationale. It is unclear whether the generated ECG can reliably reflect arrhythmias.”
>
> We appreciate the reviewer’s concern and kindly refer them to our introduction, where we discuss the motivation (lines 37-48) and applications (lines 70-73) of the conversion. Our ultimate goal is to enable accurate ECG monitoring during daily-life activities. By adopting our approach, professionals can utilize these signals (see Section 4.3) to enhance reliability, explainability, and improve decision-making in PPG-based cardiovascular classification. We have compared our approach with SOTA baselines and classification strategies, demonstrating superiority in both signal quality and classification performance. Furthermore, our uncertainty quantification measure shows that our approach achieves more reliable classification performance than the baselines (see Figures 4 and 10b). For example, Figure 4 shows that at 50% coverage, our method achieves nearly 0% classification error for “sinus arrhythmia”, while baselines show ~2.5% error. This highlights our method's reliability when high-confidence signals are selected.
>
> W.3. “The authors should further compare their classification results with state-of-the-art models”
>
> We thank the reviewer for raising this concern and agree with the suggested approach. Unfortunately, SOTA ECG classification models maintain 12-lead ECG signals, while our evaluation setup in the signal domain is restricted to lead II alone for a fair comparison with SOTA conversion methods. Importantly, the classification performance of our approach relies on the classifier’s performance (see definition of ESC 3.2). Thus, enhancing the classifier or adopting a more robust one would improve our results further compared to other baselines. Mathematical justification for this is provided in Appendix A.
>
> Q.1. “The performance of the PPG-derived ECG significantly trails that of the original ECG. How do you explain this discrepancy?”
>
> We thank the reviewer for this important question. The performance of the original ECG signal represents an upper bound for classification performance, including our own method. Intuitively, this discrepancy arises because cardiovascular labels are directly linked to heart conditions determined by ECG signals, while PPG signals only capture blood volume fluctuations, which provide partial information about the heart. As a result, PPG signals are a degraded version of ECG signals, leading to a natural decline in performance due to the loss of information. Even when generating ECG candidates from PPG signals, some information loss is unavoidable, which further reduces performance. Our approach mitigates this issue by accounting for conversion uncertainty, yielding superior results compared to other PPG-based strategies. We’ve added these details in a new appendix section (see Appendix I), and appreciate the valuable feedback of the reviewer.
>
> Q.2. “It would be beneficial to include results for other types of diseases to provide a broader evaluation.”
>
> We thank the reviewer for highlighting this important issue. While such data is not publicly available, future studies will require additional data collection to address this task. As noted in Section 6, further research involving real PPG data and associated labels is essential for a comprehensive evaluation of classification performance. However, we kindly refer the reviewer to Appendices G.2 and G.3, where additional experiments validate our setup. Appendix G.2 demonstrates high-quality synthetic PPG signal generation, and Appendix G.3 shows neglectable degradation in synthetic ECG signals derived from synthetic PPG data.
>
> Q.3. “How does pacing rhythm manifest in PPG signals?”
>
> This is a great question! A pacemaker works by sending electrical signals to the heart, triggering its contractions. However, if we use a PPG sensor to measure heart activity, it will not directly indicate that these signals are coming from a pacemaker. To identify this, we need to look for a very steady and consistent rhythm in the pulse, without the slight variations in timing that naturally occur in a healthy heart. From a more data-science perspective, we have the labeled data and we are training for predicting this condition (and others). If this prediction is hard to acquire, the classification performance will manifest this, as indeed shown in Figure 3.

---

> > ### Comment · Reviewer_r8zb · 2024-11-29
> > **Follow up discussion**
> >
> > Thank you for the detailed response.
> >
> > W2:
> > The explanation provided by the authors is not convincing. If the goal is merely to identify sinus arrhythmias, which typically reflect changes in heart rate rhythm, this can usually be achieved using simple PPG envelopes. Why is it necessary to generate ECG signals in this case? Can other types of arrhythmias be reflected in PPG signals? If they cannot, how can the ECG generated from PPG be convincing? The authors should focus on arrhythmias that are difficult to detect directly from PPG and demonstrate the feasibility of generating ECG from PPG for arrhythmia diagnosis.
> >
> > Q1:
> > The authors stated that “Even when generating ECG candidates from PPG signals, some information loss is unavoidable.” In fact, if PPG does not contain disease-related information, the generated ECG cannot include this information either. Conversely, if PPG does contain this information, it should be possible to train an effective model directly from PPG. Therefore, the rationale for solving the problem by generating ECG requires further justification.

---

> > > ### Author Response · Authors · 2024-11-29
> > >
> > > We thank the reviewer for initiating further discussion on our revision and appreciate the valuable comments provided. We strongly believe that there must have been misunderstanding, and we will do our best to clarify our rationale further.
> > >
> > > W2: “The explanation provided by the authors is not convincing. If the goal is merely to identify sinus arrhythmias, which typically reflect changes in heart rate rhythm, this can usually be achieved using simple PPG envelopes. Why is it necessary to generate ECG signals in this case?”
> > >
> > > We emphasize that our primary goal in this work is to provide accurate and more reliable PPG-based ECG signals for Holter monitoring without requiring specialized, unavailable equipment. We successfully achieved this goal through experimental validation, with the most effective currently available method for the PPG-to-ECG conversion task. The rationale behind this conversion is straightforward: in clinical settings, classification alone is insufficient for reliable predictions, as medical professionals need ECG signals to explain and verify the classification. If our method proves to be the state of the art in generating these signals, it holds significant value for this purpose.
> > >
> > > W2: “Can other types of arrhythmias be reflected in PPG signals? If they cannot, how can the ECG generated from PPG be convincing? The authors should focus on arrhythmias that are difficult to detect directly from PPG and demonstrate the feasibility of generating ECG from PPG for arrhythmia diagnosis.”
> > >
> > > Other types of heart conditions can indeed be reflected in PPG signals, as indicated by our experimental results showing classification performance across various conditions. We agree with the reviewer's suggestion regarding potential future directions for our work.
> > >
> > > Q1: We thank the reviewer for this comment. PPG signals measure blood flow fluctuations that are indeed connected to heart conditions. Verified with a cardiologist we have collaborated with, the cardiovascular information contained in PPG signals is merely partial and not dichotomous. Consequently, some heart conditions can be detected more accurately than others, although some degree of information loss is unavoidable.

---

### Official Review · Reviewer_gAqB · 2024-11-02

**Soundness:** 3
**Presentation:** 3
**Contribution:** 3
**Rating:** 6
**Confidence:** 3

**Summary:**

This paper presents a methodology to convert PPG signals into ECG ones by accounting for the uncertainty of the conversion process using a diffusion-based model. The methodology is novel arguing that by using a multi-solution approach - where multiple ECG signals can generate the same PPG wave - the combined classification of the generated ECGs is more accurate in comparison to using only the PPG or using a single generated ECG from state-of-the-art methods. The authors use two datasets, one with pairs of unlabeled ECG and PPG signals and another with only labeled ECG signals. A reverse ECG-2-PPG model is used to generate synthetic PPG for classification.

**Strengths:**

1. This work uses a diffusion model to propose multiple generated ECG waves for a single PPG signal, instead of a single solution as in previous works.
2. The paper provides the mathematical proof of the optimality of the proposed classifier, which reassures the credibility of the approach. Also, multiple performance metrics are used to evaluate the model.
3. In general the paper is well structured, with relevant tables and figures for comparison, and explanations are thorough.
4. The use of ECG-generated waves from PPG and its improved classification accuracy could be extended to the current PPG-based widespread solutions for cardiovascular monitoring.

**Weaknesses:**

Despite the comprehensive methodology and appendix, some points require some clarification:
1. The authors use 3 random seeds to generate the ECG signals from the PPG ones and report the mean and standard deviation for the chosen metrics. Given the stochastic nature of the diffusion model, are 3 seeds enough to capture the full range of ECG variability, or to provide significant statistical measures?
2. Although the motivation for the use of PPG-2-ECG relies on the widespread of wearable devices, no database with signals acquired in wearable settings was used. For the diffusion model, the MIMIC-III data comes from hospital facilities (to my knowledge), and for classification, this is unclear (unless the CinC2017 database for AFib was used, for example). This should be stated as a limitation of the work, or as a pointer for future work.

**Questions:**

1. Given the stochastic nature of the diffusion model, are 3 seeds enough to capture the full range of ECG variability, or to provide significant statistical measures in your work?
2. Although the paper reports the classification of the mean and random ECG diffusion-based solutions, you do not mention the performance of CardioGAN and RDDM directly, which were initially used to benchmark the ECG generation performance. If the end goal is to improve classification with ECG-generated signals, don't you think that it would be appropriate to use the exact implementations of CardioGAN and RDDM to compare the performance?
3. Did you ensure that there are no segments from the same recording in both the training and test datasets? I believe this is not mentioned. If the same recording appears in both datasets, the generalization ability of the approach might be not properly evaluated.

Also some missing information:
- The exact database(s) of CinC that was(were) used, as there are many databases from this source available on PhysioNet
- A data summary with the number of samples used to train and test the PPG conversion and classification models (and the corresponding class distributions)
- Only the AURC is used to assess the classification performance, but since the pathological labels often suffer from class imbalance, more insightful metrics such as sensitivity, specificity and F1-Score could be reported to improve the clinical relevance of the approach and facilitate comparison with other works.

---

> ### Author Response · Authors · 2024-11-22
>
> W.2. “... no database with signals acquired in wearable settings was used… and for classification, this is unclear... This should be stated as a limitation of the work, or as a pointer for future work.”
>
> We thank the reviewer for raising this issue. Since such data is not currently publicly available, we have revised the paper to incorporate the suggestion, adding this as a potential area for future work in the concluding remarks (see lines 537-539).
>
> W.1/Q.1. “Given the stochastic nature of the diffusion model, are 3 seeds enough to capture the full range of ECG variability, or to provide significant statistical measures in your work?”
>
> We thank the reviewer for this question. We assume the reviewer is referring to the signal quality analysis of our diffusion model presented in Table 1. We would like to clarify that we report the mean metric along with its standard error, rather than the standard deviation. According to the central limit theorem, the mean of any metric tends to follow a normal distribution if the sample size is sufficiently large. The standard error provides a 95% confidence interval for the mean estimate under the assumption of normal distribution. This makes the standard error a valuable indicator for significance in experimental comparisons. Nevertheless, to address the reviewer’s concern, we conducted additional experiments using 6 different seeds in total. The results, as shown below, demonstrate approximately similar confidence intervals around the mean, reinforcing the robustness of our findings.
>
> ----------------------------- 1-FD --------------------- 100-FD
>
> 3-seeds -------- 0.3198 ± 0.0020 -------- 0.2379 ± 0.0005
>
> 6-seeds -------- 0.3162 ± 0.0026 -------- 0.2378 ± 0.0005
>
> Q.2. “... you do not mention the performance of CardioGAN and RDDM directly, which were initially used to benchmark the ECG generation performance. If the end goal is to improve classification with ECG-generated signals, don't you think that it would be appropriate to use the exact implementations of CardioGAN and RDDM to compare the performance?”
>
> We thank the reviewer for this question and fully agree with the proposed suggestion. Unfortunately, neither RDDM nor CardioGAN provides publicly available code, which limits our ability to generate ECG signals using their methods. Therefore, the performance metrics reported in our work were taken directly from their respective papers while we ensured the same experimental setup. However, CardioGAN and RDDM are comparable to the evaluated classification strategies, specifically the combination of “Synthesized ECG: SSC Mean” and “Synthesized ECG: SSC Random” (see Appendix F), as both rely on a single random ECG solution that tends toward the mean due to mode collapse (see signal quality results in Table 1). We have revised the paper to include this clarification in Appendix F (see lines 1162-1168).
>
> Q.3. “Did you ensure that there are no segments from the same recording in both the training and test datasets?”
>
> All experiments across datasets and models were conducted on unseen test samples, which were separated from the training data by splitting records of distinct patients. We revised the paper accordingly to include this detail in Appendix E (see lines 1120-1123). Thank you for the contribution.
>
> As for the missing information,
> We thank the reviewer again for their valuable suggestions, we have revised the paper to include the details raised in these comments. Specifically,
> - For the databases of CinC - see Appendix E lines 1005-1010.
> - For data summary of both the MIMIC-III and CinC databases - see Appendices E.1.1 and E.1.2
> - For sample number and train-validation splits details - see lines 1045-1051 and lines 1020-1023.
> - For class distribution report - see Figure 8.
> - For metrics evaluating performance considering the imbalance - see Appendix H.2.

---

### Official Review · Reviewer_4rAh · 2024-11-04

**Soundness:** 4
**Presentation:** 2
**Contribution:** 3
**Rating:** 6
**Confidence:** 5

**Summary:**

This paper presents "Uncertainty-Aware PPG-2-ECG (UA-P2E)," a novel framework that employs diffusion models to convert photoplethysmography (PPG) signals into electrocardiography (ECG) signals for improved cardiovascular disease classification. Recognizing the ill-posed and inherently ambiguous nature of the PPG-to-ECG conversion—stemming from the loss of certain physiological information in PPG measurements—the authors propose a multi-solution approach to address this challenge. By leveraging diffusion models, UA-P2E captures the full distribution of possible ECG signals corresponding to a given PPG input, effectively modeling the uncertainty inherent in this inverse problem. This allows the framework to generate robust ECG signals that account for the variability and ambiguity of the conversion process. The authors validate their approach through experiments across multiple cardiovascular conditions, demonstrating state-of-the-art classification performance. They provide empirical evaluations, including comparisons with two baseline models, to substantiate the effectiveness of UA-P2E in both signal reconstruction and cardiovascular classification tasks.

**Strengths:**

Originality and Significance: This paper offers a novel approach to the challenging PPG-to-ECG conversion by applying a diffusion-based, uncertainty-aware model. This method effectively addresses the inherently ill-posed nature of the task, capturing the distribution of possible ECG outputs rather than a single solution. By doing so, it meets a main need in cardiovascular diagnostics, especially where paired data is limited.

Methodological Rigor: The paper demonstrates strong methodological rigor with a solid theoretical foundation, including proofs of the Expected Score Classifier's (ESC) optimality. This rigorous analysis, supported by detailed equations (e.g., Theorem 3.1), supports the model’s reliability, ensuring both empirical soundness and theoretical robustness.

Comprehensive Evaluation: A thorough evaluation across 11 cardiovascular conditions highlights the model's generalizability and robustness. Compared to baseline models, it shows superior performance in signal reconstruction and classification, with added metrics for uncertainty quantification. This detailed analysis strengthens the case for clinical applicability.

Clarity and Presentation: The paper is well-organized, balancing technical details with intuitive explanations, such as figures illustrating model performance and ECG visualization strategies, enhancing clarity. The emphasis on interpretability supports practical use in clinical settings.

Significance for the Field: The integration of uncertainty-aware diffusion models for physiological signal conversion represents a meaningful enhancement in machine learning for healthcare. The interdisciplinary approach bridges machine learning and biomedical engineering with the potential to drive future innovations in cardiovascular diagnostics and medical device development.

**Weaknesses:**

Dependence on Synthetic Data: The evaluation heavily relies on synthetic PPG data, especially for augmenting the CinC dataset (Section 5.2​). This dependence raises concerns about potential biases, as synthetic data may not fully capture the variability and complexities of real-world PPG signals. Consequently, the model's generalizability to real clinical settings might be limited, potentially affecting its practical applicability.

Limited Baseline Comparisons: The paper compares UA-P2E primarily with two baseline models: CardioGAN and RDDM (Table 1). While these comparisons provide some insight, the limited scope restricts a comprehensive understanding of the model's performance relative to the broader range of existing methodologies. Including additional, especially more recent baseline models, such as those in the referenced ArXiv papers, would strengthen the evaluation and better position UA-P2E within the current state-of-the-art.

Potential for Synthetic Artifacts: The possibility of generating hallucinations or artifacts in the synthetic ECG signals produced by the diffusion models is not thoroughly examined. Since diffusion-based models can introduce unrealistic features, a lack of analysis on this front may raise concerns about the reliability and clinical validity of the generated signals. Addressing this risk through quantitative assessments would enhance the credibility of the proposed approach.

Limited Dataset Diversity: The study focuses on paired PPG-ECG datasets from CinC and MIMIC-III. This narrow dataset selection may not adequately demonstrate the model's flexibility or adaptability to other data sources. Expanding the evaluation to include larger or unpaired datasets would provide a more robust validation of the model's generalizability and its potential utility across diverse cardiovascular data.

**Questions:**

Hallucination Analysis: How does the model ensure that synthetic ECG signals do not contain unrealistic artifacts?

Baseline Comparisons: The comparison is limited to CardioGAN and RDDM. Have you considered including additional baseline models from the literature (e.g., ArXiv:2309.15375, 2012.04949, 2204.11795, 2101.02362) to provide a more comprehensive performance evaluation?

Real vs. Synthetic Data Performance: How do the model's performance metrics change when trained on real versus synthetic PPG-ECG pairs? Can you quantify the impact of synthetic data on classification accuracy?

Generalizability Across Datasets: Have you considered applying this approach to unpaired PPG-ECG datasets or datasets from different demographic groups? Would such testing be feasible within your current framework? Have you employed techniques such as cross-validation or tested on external datasets to ensure that your model generalizes well beyond the training data?

Risk of Overfitting: Given the complexity of diffusion models and the size of the datasets used, what measures have you taken to prevent overfitting?

---

> ### Author Response · Authors · 2024-11-22
>
> W.1. Dependence on Synthetic Data
>
> We thank the reviewer for the comment. Our analysis of the MIMIC database used real clinical PPG and ECG signals, while synthetic PPG data was used for the CinC database evaluation. In Appendix G, we analyze the quality and classification performance of synthetic PPG data. We show high-quality synthetic PPG generation (Appendix G.2) with neglectable degradation in derived synthetic ECG signals (Appendix G.3). Afib classification trends align with prior experiments (Appendix G.1), demonstrating that our multi-solution approach outperforms baselines and that real and synthetic PPG data perform similarly. However, as noted in Section 6, further research with real PPG data and labels is needed, necessitating future data collection due to the lack of publicly available datasets.
>
> W.3/Q.1. Potential for Synthetic Artifacts
>
> True, we acknowledge the risk of generating hallucinations and partially address this in our paper. Section 4.2 introduces an ESC-based selection process ensuring reliability by rejecting low-performing PPG signals, as validated by Risk-Coverage curves. Appendix B provides details on a calibration scheme to ensure reliability for unseen i.i.d. data. This approach is validated in our experiments (Figures 4 and 10b) with Risk-Coverage curves showing that our multi-solution method (green) achieves greater reliability. Lastly, we do note in Section 6, further research is necessary to enhance reliability concerning artifacts and hallucinations in the signal domain.
>
> W.4. Limited Dataset Diversity
>
> We appreciate the reviewer’s concern and apologize for any misunderstanding. Our experiments utilized the MIMIC-III and CinC databases. The MIMIC-III database is a thorough and public dataset of paired PPG/ECG signals from diverse patient populations, used to assess signal quality. However, its lack of cardiovascular labels limits its use for classification. Therefore, we evaluated our approach using CinC, which is the largest and most challenging public dataset for cardiovascular classification, including various conditions across diverse populations. We believe that these datasets represent the most challenging benchmarks for the PPG-2-ECG conversion.
>
> Q.2/W.2. Baseline Comparisons
>
> We thank the reviewer for highlighting this concern. Unfortunately, none of the relevant publications provide code for their work. We selected RDDM as the most recent and relevant diffusion-based baseline (currently SOTA) and CardioGAN to represent GAN-based approaches. Most PPG-to-ECG studies rely on MAE/MSE loss, which generates average signals rather than high-quality outputs [1]. Other generative methods often use Variational Inference with Gaussian assumptions or GANs. Our approach avoids Gaussian assumptions, offering greater robustness, while GANs, prone to mode collapse, lack output variability, as evidenced by CardioGAN's results in Table 1.
>
> [1] Blau et al. "The perception-distortion tradeoff." 2018.
>
> Q.3. Real vs. Synthetic Data Performance
>
> We kindly refer the reviewer to Appendix G.1, where Figure 10 illustrates the classification impact and compares real and synthetic PPG-ECG pairs (lines 1212-1217). Nonetheless, it is evident that similar trends are observed across all reported results (see Figures 3, 4 and 10), further reinforcing the validity of our experiments.
>
> Q.4. Generalizability Across Datasets
>
> Thank you for your valuable questions. Extending our model to support unpaired PPG-ECG signals is an interesting direction for future work. Our model aligns with the timesteps of the provided PPG signals, so it cannot currently predict ECG at unobserved timesteps. We have not tested specific demographic groups, as our aim was broad generality. The MIMIC-III dataset, used for training, is the largest available and includes diverse demographics. For cross-validation, all experiments were conducted with 3 seeds, each having a unique train-test split through distinct patients. These details were added in Appendix E.2 (see lines 1056-1057) and in Appendix E.3 (see lines 1120-1123) of the revised paper.
>
> Q.5. Risk of Overfitting
>
> To address overfitting, we implemented noise-cleaning preprocessing to prevent the model from learning noise present in the training data (see Appendix E.1.1). Additionally, diffusion models inherently benefit from implicit regularization due to their robust mathematical framework, which is well-suited for mitigating overfitting. However, we have carefully addressed the reviewer’s concern, by conducting further analysis and comparing the performance of our denoising model on training and test data. Furthermore, we performed an experiment that included a histogram of distances between ECG signals, comparing the true ECG to its nearest ECG sample. This analysis demonstrates how frequently the true ECG can be approximated. We have included a new appendix section, containing the analysis (see Appendix D), and appreciate the valuable feedback of the reviewer.

---

### Author Response · Authors · 2024-11-22

We thank the reviewers for their constructive feedback and for recognizing the performance and novelty of our method. We are committed to thoroughly address all concerns raised by the reviewers and to conduct additional experiments to strengthen our presentation. The additional experiments further validate our methodology and enhance our experimental results. We welcome further discussion and insights from the reviewers.

---

### Author Response · Authors · 2024-11-24

Dear reviewers,

The discussion stage is approaching its end, but it hasn't been as active as we had hoped. We would greatly appreciate it if you could take the time to address our response.

Thanks,

The authors.

---

### Meta-Review · Area_Chair_TcGd · 2024-12-19

**Metareview:**

This paper presents an approach to generate ECG from PPG for cardiovascular disease detection. However, although the PPG and ECG signals are inter-related, there is a lack of strong evidence that we can use PPG to detect cardio diseases. As pointed out by the authors, the conversion from PPG to ECG is ill-posed problem. Then the question comes: how do we know the correct ECG is obtained (even if we assume that PPG contain full information of the heart diseases). The reviewers have the concerns on the motivation of the paper as PPG cannot solve some of problems and therefore PPG generated ECG cannot solve the problem too. There is also a concern on the lack of test in real-world data and/or the reliability of such method. I agree with the reviewers there is a lack of motivation to generate ECG from PPG. Why not use PPG directly?

Overall, I cannot recommend to accept this paper.

**Additional Comments On Reviewer Discussion:**

The reviewers have some major concerns on the paper:
1) the motivation for the paper—using PPG to generate ECG—does not make sense. It does not seem feasible to accurately generate ECG signals with pacing rhythm characteristics from PPG. Data from real-world test is needed to support this, which is inline with the second major weakness.
2) the lack of test in real-world data and reliance on the diverse of the synthetic data
I agree with the reviewer that we can not generate ECG signals from PPG to address problems that PPG itself can solve. This concern makes the second issue very important: the validation with real-world data.

---

### Decision · Program_Chairs · 2025-01-22

Reject